# Displacement efficiency of alternative energy and trans-provincial imported electricity in China

Yuanan Hu[1] & Hefa Cheng[2]

China has invested heavily on alternative energy, but the effectiveness of such energy sources at substituting the dominant coal-fired generation remains unknown. Here we analyse the displacement of fossil-fuel-generated electricity by alternative energy, primarily hydropower, and by trans-provincial imported electricity in China between 1995 and 2014 using two-way fixed-effects panel regression models. Nationwide, each unit of alternative energy displaces nearly one-quarter of a unit of fossil-fuel-generated electricity, while each unit of imported electricity (regardless of the generation source) displaces $\sim 0.3$ unit of fossil-fuel electricity generated locally. Results from the six regional grids indicate that significant displacement of fossil-fuel-generated electricity occurs once the share of alternative energy in the electricity supply mix exceeds $\sim 10\%$, which is accompanied by 10–50% rebound in the consumption of fossil-fuel-generated electricity. These findings indicate the need for a policy that integrates carbon taxation, alternative energy and energy efficiency to facilitate China's transition towards a low-carbon economy.

[1] MOE Laboratory of Groundwater Circulation and Evolution, School of Water Resources and Environment, China University of Geosciences (Beijing), Beijing 100083, China. [2] MOE Laboratory for Earth Surface Processes, College of Urban and Environmental Sciences, Peking University, Beijing 100871, China. Correspondence and requests for materials should be addressed to H.C. (email: hefac@umich.edu).

China accounts for over 16% of the gross domestic product (GDP), 23% of the energy consumption and 17% of the renewable energy production of the world[1]. It is also the world's largest consumer of electricity (5,638 TWh in 2014), with thermal generation accounting for over three-quarters of the total supply[2]. Due to the limited domestic production of oil and natural gas, thermal power generation is fuelled predominantly by coal (89.0%)[3,4]. As a result, China is the world's biggest emitter of anthropogenic greenhouse gases (GHGs), and coal-fired power generation contributes largely to the widespread air pollution[3,5].

Increasing the production of alternative energy to substitute fossil fuels has been pursued worldwide to improve the sustainability and mitigate the environmental impact of energy use[6–8]. China has also significantly expanded the investment on renewable energy sources and nuclear power, and stepped up the construction of transmission networks to connect the renewable electricity produced[4,5,9]. Hydropower and nuclear power accounted for 18.8 and 2.3% of China's electricity production, respectively, while the contribution from solar, wind, biomass and geothermal combined merely reached 2.8% by the end of 2014 (ref. 2). Already the world's largest producer of hydroelectricity, wind energy and solar power, renewable energy generation in China grew by 20.9% in 2015 (ref. 1). Meanwhile, nuclear power in China is growing quickly, despite of the phase-out in some European countries after the recent Fukushima nuclear disaster[10]. The rapid and significant change in China's energy structure not only shapes a large part of the global energy consumption and GHG emissions, but also has important influence on the economic growth worldwide.

In the most ideal scenario, each unit of electricity supplied by non-fossil-fuel sources avoids the generation of an equal amount of electricity by existing generators burning fossil fuels. Such proportional displacement is often tacitly assumed when developing national climate and energy policies[11,12]. However, the actual mechanism of fossil fuel displacement is rather complicated, and is influenced by a range of structural, institutional and behavioural factors[11,12]. The electric power industry has been developed from and is well accustomed to fossil-fuel-fired generation, while the infrastructure, production cost and supply characteristics of alternative energy sources can be significantly different. Based on their costs of generation and the flexibility in meeting the peak load, nuclear power and hydropower are often adopted as baseload resources, followed by coal, natural gas and oil-fired generation (Supplementary Table 1). Although wind energy and solar power typically have low variable costs, they are inherently weather-dependent and intermittent, and their supply does not follow the typical demand curve, which makes their integration into the power grid challenging[4]. Furthermore, large renewable energy sources (for example, hydro and wind) are mostly located in the remote areas, and long-distance transmission networks are required to transmit the electricity to the load centres in major cities and industrial hubs, where the electricity demand is often met by local coal plants. Approximately 6–7% of the electricity fed into the grid was lost during transmission and distribution in China (Supplementary Fig. 1), while renewable electricity generated in some remote regions was not grid-connected due to insufficient transmission capacity. It has been reported that the northwestern provinces and regions rich in wind power resources, including Inner Mongolia, accounted for three-quarters of China's unconnected wind power generation[4]. In addition, when sourcing power, state-owned electrical grid companies often give priority to the electricity generated by local coal plants, which have relatively high capital costs, over the electricity supplied from renewable sources or imported from the other provinces. Therefore, besides technological limitations, a range of practical, economic, political factors and the subjective considerations of the grid operators, can all influence the actual substitution of fossil-fuel-generated electricity by alternative energy.

A pioneering study by York found that the efficiency of non-fossil-fuel energy sources and the electricity generated from them was less than one-quarter and one-tenth for substituting fossil fuels and fossil-fuel-generated electricity, respectively, in 132 nations/regions of the world over the 1960–2009 period[11,12]. Contradictory to the presupposed proportional displacement, more than 11 kWh of non-fossil-fuel electricity was required to displace 1 kWh of fossil-fuel-generated electricity[11,12]. As the reduction in fossil fuel consumption was far from being commensurate with the increase in alternative energy production, the growth of alternative energy should be much more massive than previously thought to substantially reduce the global consumption of fossil fuels[11]. These findings also highlight the importance of changing the political, economic and cultural contexts to facilitate the displacement of fossil fuels and to curb the energy consumption at the individual/household and system levels during the course of renewable energy development[11,12].

Although China has become one of the world's largest renewable energy markets, the efficiency of alternative energy sources for replacing fossil fuels is unknown. In addition, no previous research has investigated the use efficiency of trans-provincial transported electricity, which plays a vital role in balancing the significant spatial mismatch of power generation and consumption in the country (Fig. 1). In this work, we used two-way fixed-effects models to determine the efficiency at which alternative energy and trans-provincial imported electricity (regardless of the source of generation) displaced the electricity generated by local fossil-fuel-fired plants (that is, within the provinces) both nationwide and in the six regional grids over the period of 1995–2014. The results show that each unit of alternative energy displaces 0.231 ± 0.078 unit of fossil-fuel-generated electricity in China, which is twice of the global average level, while each unit of imported electricity substitutes 0.312 ± 0.108 unit of locally generated electricity. Alternative energy, primarily hydropower, can have significant displacement effect on fossil-fuel-fired generation only when its share in the electricity supply mix is increased above 10% or more in the country's six inter-provincial regional power grids. The findings provide the very first estimation on the displacement effect of alternative energy and trans-provincial imported electricity on local fossil-fuel-fired electricity generation in China, a key player on the global energy market. They also reveal important insights on effective policies to facilitate decoupling of economic growth from fossil energy consumption and thus GHG emissions in China.

## Results

**Efficiency of alternative energy and imported electricity.** Table 1 summarizes the displacement coefficients of alternative energy and trans-provincial imported electricity from three models (see Supplementary Table 2 for full results). Model 1 reveals a displacement coefficient of − 0.231 ± 0.078 for alternative energy, indicating that only 0.231 unit of fossil-fuel-generated electricity could be substituted by each unit of electricity produced from alternative energy sources in China between 1995 and 2014. Comparable displacement coefficients ( − 0.221 ± 0.079 and − 0.241 ± 0.075) are also observed for alternative energy in models 2 and 3. These results consistently indicate that each unit of electricity produced by alternative energy sources displaced, on the average, nearly one-quarter of a unit of fossil-fuel-generated electricity. For the purpose of comparison, we also investigated

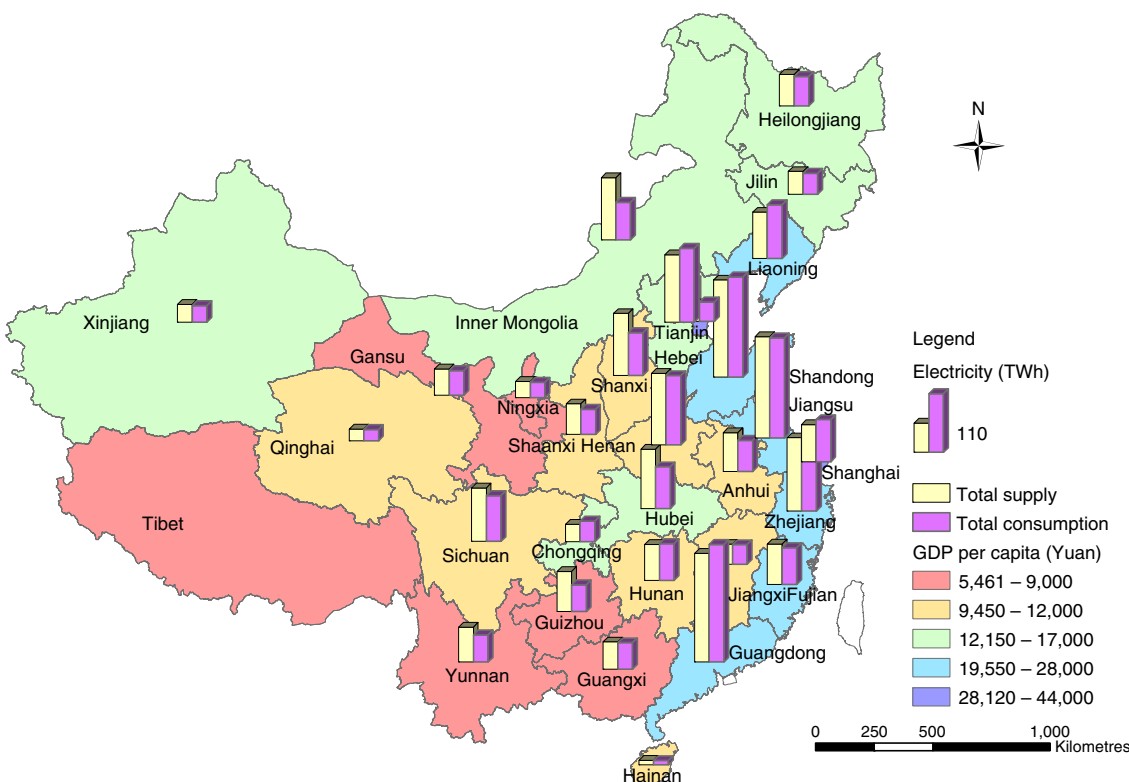

**Figure 1 | Variations in electricity use and economic conditions in China.** The map shows the electricity generation and consumption, and GDP per capita across all 30 provinces and municipalities of mainland China (excluding Tibet) in the year of 2014. Around one-third of the coal electricity in China is generated by the power plants located right near large coal mines in the northern and western provinces[4], whereas over 40% of the electricity is consumed by the major industrial and urban centres in the coastal provinces, which account for about half of the national GDP and industrial output.

**Table 1 | Displacement coefficients of alternative energy and trans-provincial imported electricity in China estimated from models of electricity supply.**

| Predictor variable | Model 1 | Model 2 | Model 3 |
|---|---|---|---|
| Alternative energy per capita | − 0.231* | − 0.221* | − 0.241* |
| | (0.078) | (0.079) | (0.075) |
| | [0.003] | [0.005] | [0.001] |
| Trans-provincial imported electricity per capita | − 0.312* | − 0.313* | − 0.235* |
| | (0.108) | (0.106) | (0.104) |
| | [0.004] | [0.003] | [0.024] |

Shown are displacement coefficients estimated from panel analyses of data from 30 provinces and municipalities during 1995–2014: electricity demand is controlled by GDP per capita in model 1, while models 2 and 3 are constructed with the addition of other variables, including the share of non-agricultural GDP and the percentage of urban population. Full results for these three models are presented in Supplementary Table 2.
*Statistically significant at the 0.05 alpha level (two-tailed test), s.e.'s are reported in parentheses, while P values are presented in brackets.

the global displacement of fossil-fuel-generated electricity by alternative energy using our model. The displacement coefficient of alternative energy between 1960 and 2009 was found to be − 0.136 ± 0.020, which is close to that (− 0.089 ± 0.009) obtained in York's cross-national longitudinal study[12], while that over the 1995–2013 period was − 0.114 ± 0.018 (Supplementary Table 3). That is, the displacement of fossil-fuel-generated electricity by alternative energy in China was two times more efficient than the global average over the past two decades. For trans-provincial imported electricity, models 1–3 reveal that its displacement coefficient ranged from − 0.313 to − 0.235 with s.e.'s of ∼0.1. Thus each unit of electricity imported from other provinces (regardless of the generation source) substituted ∼0.3 unit of fossil-fuel electricity generated locally (that is, within the province).

Instead of a unified national grid system, China has six wide area synchronous grids, although the under-developed long-distance transmission capacity has prevented efficient re-routing of electricity from the regions with surplus generation to those in shortages[4]. Table 2 compares the displacement coefficients of alternative energy and trans-provincial imported electricity in the six inter-provincial regional power grids (see Supplementary Table 4 for full results). With the exception of the north and northeast grids, which had low shares (<8%) of alternative energy in their supply mixes (Supplementary Table 5), relatively good to strong displacement effect was observed for alternative energy in the others (− 0.488 to − 0.888). The imported electricity showed no or statistically insignificant effect on substituting local fossil-fuel-fired power generation in the northwest and northeast grids, which had relatively high exportation of

**Table 2 | Displacement coefficients of alternative energy and trans-provincial imported electricity in China's six inter-provincial regional power grids.**

| Predictor variable | Inter-provincial regional power grid | | | | | |
|---|---|---|---|---|---|---|
| | East | Central | North | Northeast | Northwest | South |
| Alternative energy per capita | − 0.709* | − 0.672* | 3.638* | 3.960* | − 0.488* | − 0.888* |
| | (0.142) | (0.065) | (1.267) | (0.608) | (0.099) | (0.072) |
| | [<0.001] | [<0.001] | [0.004] | [<0.001] | [<0.001] | [<0.001] |
| Trans-provincial imported electricity per capita | − 0.197* | − 0.457* | − 0.841* | − 0.339 | 0.816* | − 0.891* |
| | (0.078) | (0.105) | (0.083) | (0.411) | (0.311) | (0.108) |
| | [0.012] | [<0.001] | [<0.001] | [0.409] | [0.009] | [<0.001] |

Shown are displacement coefficients estimated from panel analyses of data from 30 provinces and municipalities during 1995–2014, where the electricity demand is modelled as controlled by GDP per capita. Full results for the model are presented in Supplementary Table 4.
*Statistically significant at the 0.05 alpha level (two-tailed test), s.e.'s are reported in parentheses, while $P$ values are presented in brackets.

**Table 3 | Displacement coefficients of hydropower and non-hydro alternative energy sources nationwide and in the six inter-provincial regional power grids.**

| Predictor variable | Nationwide | Inter-provincial regional power grid | | | | | |
|---|---|---|---|---|---|---|---|
| | | East | Central | North | Northeast | Northwest | South |
| Hydropower per capita | − 0.637* | − 1.168* | − 0.682* | 5.937* | 1.160 | − 0.543* | − 0.896* |
| | (0.091) | (0.117) | (0.064) | (1.909) | (1.008) | (0.097) | (0.073) |
| | [<0.001] | [<0.001] | [<0.001] | [0.002] | [0.250] | [<0.001] | [<0.001] |
| Non-hydro alternative energy per capita | 2.399* | 0.064 | − 1.251 | 3.167* | 4.447* | 1.911* | − 0.615 |
| | (0.657) | (0.206) | (0.886) | (1.358) | (0.642) | (0.824) | (0.575) |
| | [<0.001] | [0.756] | [0.158] | [0.020] | [<0.001] | [0.020] | [0.285] |
| Trans-provincial imported electricity per capita† | − 0.271* | − 0.180* | − 0.464* | − 0.772* | − 0.170 | 0.532 | − 0.886* |
| | (0.106) | (0.075) | (0.105) | (0.087) | (0.347) | (0.338) | (0.109) |
| | [0.011] | [0.016] | [<0.001] | [<0.001] | [0.625] | [0.116] | [<0.001] |

Shown are displacement coefficients estimated from panel analyses of data from 30 provinces and municipalities during 1995–2014, where the electricity demand is modelled as controlled by GDP per capita. The displacement coefficients of trans-provincial imported electricity are summarized along with those of hydropower and non-hydro alternative energy sources, while full results for the model are presented in Supplementary Table 7.
*Statistically significant at the 0.05 alpha level (two-tailed test), s.e.'s are reported in parentheses, while $P$ values are presented in brackets.
†As expected, the displacement coefficients for trans-provincial imported electricity nationwide and in the six regional grids are comparable to those obtained by the previous models (Tables 1 and 2).

electricity (Supplementary Table 6). Strong displacement occurred in the south grid ($-0.891 \pm 0.108$) and north grid ($-0.841 \pm 0.083$), while the central grid ($-0.457 \pm 0.105$) and east grid ($-0.197 \pm 0.078$) exhibited much poorer displacement. Together, the above results indicate that strong displacement could be achieved by both alternative energy and inter-provincial transmission in some grids, while the national average levels were significantly reduced by the poor efficiency in the rest.

**Efficiency of hydropower and non-hydro alternative energy.** Accounting for over three-quarter of the electricity generation from non-fossil-fuel sources in China, the displacement effect of hydropower is of particular importance. Table 3 shows that the national average displacement coefficient of hydropower ($-0.637 \pm 0.091$) is almost three times that of alternative energy ($-0.231 \pm 0.078$), while no displacement effect is observed for non-hydro alternative energy (see Supplementary Table 7 for full results). The lack of significant displacement effect by non-hydro alternative energy is consistent with the fact that large-scale deployment of such energy sources barely occurred in China over the period of 1995–2014 (Supplementary Fig. 2), which is also consistent with the global substitution pattern observed in York's study[12].

As expected, the displacement coefficient of hydropower exhibited very large variation among the regional grids. Essentially, proportional displacement occurred in the east grid ($-1.168 \pm 0.117$), and strong displacement took place in the south grid ($-0.896 \pm 0.073$), followed by the central grid ($-0.682 \pm 0.064$) and northwest grid ($-0.543 \pm 0.097$). In

contrast, no statistically significant displacement was observed in the north and northeast grids, which had rather low shares ($<5\%$) of hydropower in their electricity supply mixes (Supplementary Table 8). As shown in Fig. 2, the distribution of hydropower resources in China is highly uneven, and so is the hydropower generation capacity[13]. It appears that hydropower had little displacement effect on fossil-fuel-fired generation in the provinces with low production. Non-hydro alternative energy did not show statistically significant displacement effect in any of the six regional grids, in which its mean share was at most 6.6% (Supplementary Table 8).

The displacement coefficient of hydropower was negative and significant ($-0.573 \pm 0.092$) in the provinces ranked among the top half in hydropower production per capita, whereas no displacement effect existed in the provinces ranked among the bottom half (Table 4, with full details shown in Supplementary Table 10). In contrast, the trans-provincial imported electricity had relatively good displacement effect ($-0.619 \pm 0.108$) on local fossil-fuel-fired generation in the provinces ranked among the bottom half in hydropower production per capita, but no effect in the others. These observations indicate that somehow the provinces with enough locally produced, inexpensive hydropower did not make effective use of the trans-provincial imported electricity.

**Dependence of efficiency on alternative energy penetration.** The displacement efficiency of fossil-fuel-generated electricity by alternative energy and trans-provincial imported electricity varied largely across China's six regional grids. Figure 3a depicts the

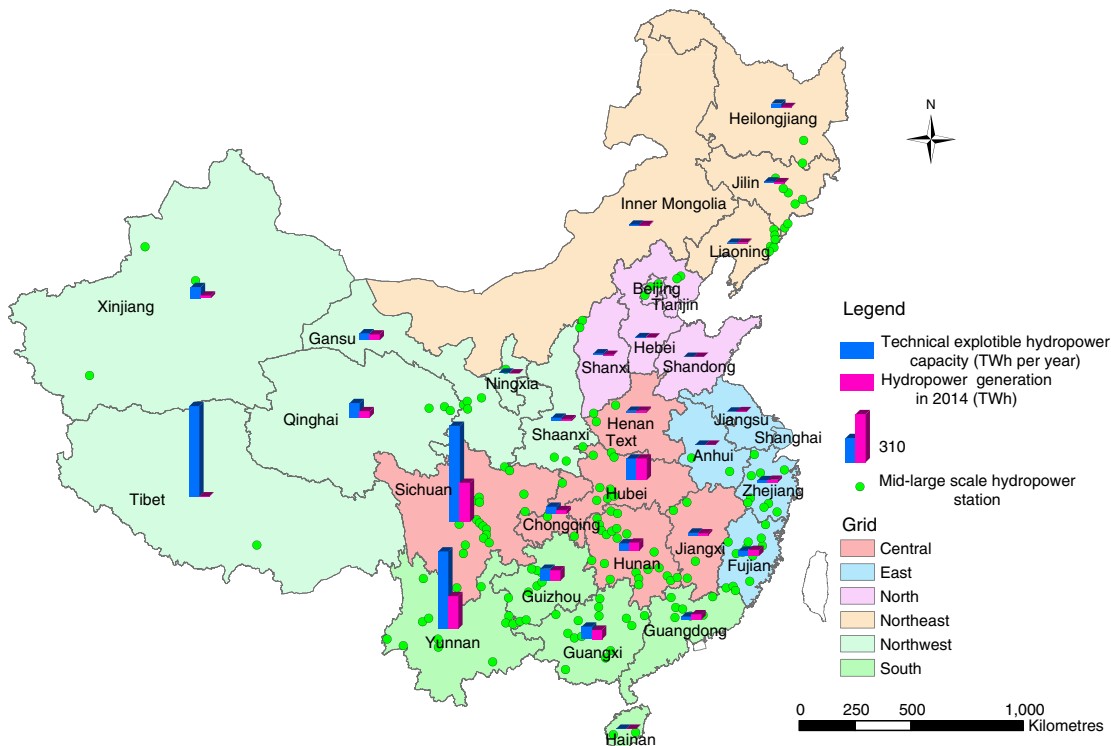

**Figure 2 | Spatial distribution of hydropower resources and production in China.** The map shows the technically exploitable hydropower resources, hydropower generation (2014) and major hydropower stations in all provinces and municipalities of mainland China. The boundaries of the six inter-provincial regional power grids are also shown. Most of the hydropower bases are located in the central and southern provinces, including Sichuan and Yunnan, while the northeastern and northern provinces have very low hydropower generation.

**Table 4 | Displacement coefficients of hydropower and non-hydro alternative energy sources in the provinces ranked among the top and bottom halves in hydropower production per capita.**

| Predictor variable | Provinces with hydropower production per capita ranked in[*] | |
| --- | --- | --- |
| | **Top half** | **Bottom half** |
| Hydropower per capita | − 0.573[†] | 1.133 |
| | (0.092) | (0.901) |
| | [<0.001] | [0.209] |
| Non-hydro alternative energy per capita | 2.684[†] | 2.646* |
| | (0.673) | (0.671) |
| | [<0.001] | [<0.001] |
| Trans-provincial imported electricity per capita | − 0.049 | − 0.619[†] |
| | (0.162) | (0.108) |
| | [0.764] | [<0.001] |

Shown are displacement coefficients estimated from panel analyses of data from 30 provinces and municipalities during 1995–2014, where the electricity demand is modelled as controlled by GDP per capita. The displacement coefficients of trans-provincial imported electricity are summarized along with those of hydropower and non-hydro alternative energy sources, while full results for the model are presented in Supplementary Table 10.
*Overall ranking of the 30 provinces and municipalities in hydropower production per capita during 1995–2014 is listed in Supplementary Table 9.
†Statistically significant at the 0.05 alpha level (two-tailed test), s.e.'s are reported in parentheses, while P values are presented in brackets.

relationship between the displacement coefficient of alternative energy and its share in the grid's electricity supply mix. Alternative energy exhibited no displacement effect on fossil-fuel-generated electricity when its share in the grid's supply mix was very low, while relatively good to strong displacement (− 0.488 to − 0.888) occurred in the grids with shares above 10%.

Figure 3b,c shows the relationship between the displacement coefficients of hydropower and non-hydro alternative energy and their shares in the grid's electricity supply mix, respectively. As expected, the displacement effect of hydropower resembles that of alternative energy because of its overwhelming contribution. In contrast, non-hydro alternative energy showed no statistically

significant displacement effect on fossil-fuel-generated electricity. Similarly, the displacement coefficient of trans-provincial imported electricity had no apparent dependence on the share of imported electricity or that of net inflow in the grid's electricity supply mix (Supplementary Fig. 3). These results consistently suggest that significant penetration of alternative energy on the grid might be necessary to achieve meaningful displacement of fossil-fuel-generated electricity. For imported electricity, its use efficiency depended primarily on the region, and good displacement could occur in both regions with net inflow of electricity (for example, north grid) and those with net outflow of electricity (for example, central grid).

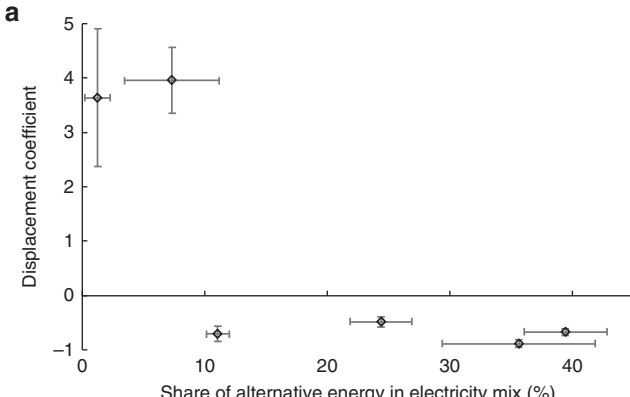

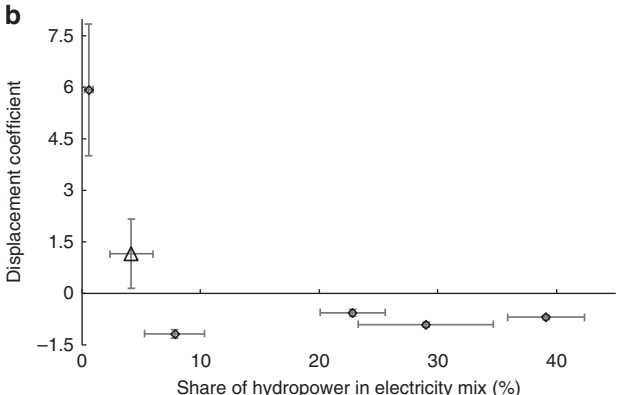

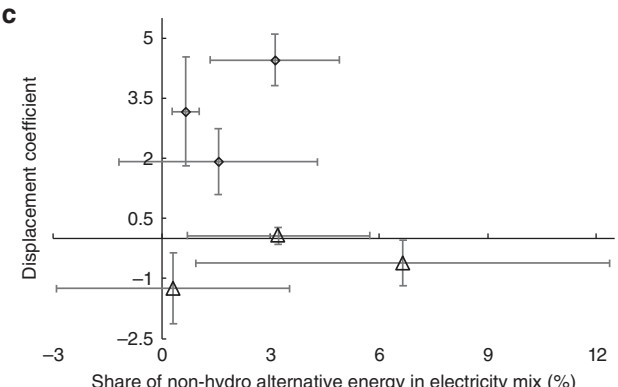

**Figure 3 | Dependence of alternative energy's displacement effect on its grid penetration.** Shown are the displacement coefficients of (**a**) alternative energy, (**b**) hydropower and (**c**) non-hydro alternative energy for substituting fossil-fuel-generated electricity by local plants (within the provinces) as a function of its penetration in the electricity supply mix of China's six inter-provincial regional power grids. Error bars represent s.e.m., and statistically significant and insignificant displacement coefficients (compared to 0, $P$ value $<0.05$, two-tailed test) are plotted with the symbols of ''diamond'' and ''triangle'', respectively.

## Discussion

The results presented above indicate that alternative energy in China displaced fossil-fuel-fired electricity generation at twice the efficiency of the global average. This is mainly attributed to the unquenchable thirst for electric power during the course of fast industrialization and urbanization in China. The displacement efficiency of alternative energy can be largely different among countries due to the facts that the growth, technology and consumption pattern vary significantly, and that the energy consumption of a country evolves as it goes through the stages of

economic growth[14]. The availability, efficiency and costs of energy are a significant constraint to industrial activities in the developing countries, particular those with unreliable supply[15]. Regional and seasonal power shortages had emerged in the coastal provinces of China since the early 2000s (ref. 4), and with the existence of such significant unmet demand, the increased supply of alternative energy would be readily consumed in growing the economy, resulting in relatively high overall efficiency for displacing fossil-fuel-fired generation.

The heavy dominance of hydropower, which cost-effectively and readily substitutes coal-fired generation at supplying base-load[4], in alternative energy production probably also contributed to the relatively good displacement effect observed in China. Essentially, proportional displacement of fossil-fuel-generated electricity by hydropower occurred in the east grid, which suffered chronic power shortages and had significant production of hydropower (averaged at 7.8% of the supply mix). In contrast, in the provinces poorly endowed with hydropower resources, the limited hydropower was typically generated from mid- and small-sized facilities and used primarily for rural electrification[4], thus could contribute little to local industrial growth. Similarly, with low mean shares in the supply mixes, non-hydro alternative energy sources could not achieve meaningful displacement effect on fossil-fuel-fired generation in any of the regional grids.

With a highly centralized government in China, energy policies, government and industry regulations, fuel prices, government subsidies and tariffs and grid management are relatively uniform across all six regional grids. Alternative energy was found to displace fossil-fuel-generated electricity at efficiency in the range of 50–90% only when its share in the grid's supply mix exceeded $\sim 10\%$. Although the correlation observed across different grids is not necessarily causation, it is strongly suggestive that increasing the share of alternative energy beyond a certain threshold ($\sim 10\%$) in the electricity supply mix is necessary to achieve significant displacement effect. This could result from the presence of economies of scale and learning effects in alternative energy development[16], that is, with increased output of alternative energy, the average cost of production decreased while the use efficiency also became greater across the grid.

Among the six regional grids, only the north and east grids had net inflow of electricity with mean shares of 5.4 and 3.8% over the period of 1995–2014, respectively (Supplementary Table 6). The displacement of local fossil-fuel-generated electricity by trans-provincial imported electricity depended strongly on the local supply: the regions suffering chronic power shortages (that is, the south and north grids) had high efficiency while those with surplus generation (that is, the northwest and northeast grids) had low efficiency. These results indicate that exogenous electricity (generated by alternative energy sources locally or imported from other provinces) could be utilized more efficiently in the regions with significant unmet demand. This calls for significant improvement on energy efficiency in the provinces with surplus generation and expansion of the long-distance transmission networks to re-route the surplus electricity more efficiently across the regional grids.

In addition to factors such as transmission and distribution losses and under-utilization, the inefficient substitution of fossil-fuel-generated electricity by alternative energy could also be contributed by the consumption behaviours of those that demand energy. The Jevons Paradox, that is, increasing the productivity of a commodity leads to greater consumption, has long been documented in energy economics and policy research[14,17,18]. The savings achieved from improved energy efficiency can be partially or even fully cancelled out due to greater consumption of energy stimulated by the lower cost of energy services[18–20]. A substantial amount of empirical evidence supports the existence of such

rebound effect, which can be considered as a behavioural or other systemic response to improved energy efficiency[14,15,18,20–22]. Nonetheless, the magnitude of the rebound effect is limited, typically offsetting 20–60% of the expected energy savings[14,21,23]. Although very little research has investigated the relationship between renewable energy generation and energy consumption, an exogenous increase in the supply of energy from non-fossil-fuel sources, can also induce greater energy consumption, that is, rebound effect, in a way similar to that of energy efficiency[24,25]. Studies have suggested that the ethanol subsidy in the United States could lower the price of E10 (10% ethanol and 90% gasoline), which might eventually result in greater gasoline demand and more oil imports[26,27]. Rebound effect was also observed for residential electricity consumption in the states adopting one or more financial incentives for renewable energy in the United States[28].

Similar to the case of efficiency improvement, with technological advances in alternative energy generation and expansion of the production capacity, the costs of energy services (with or without subsidies) are reduced, which can result in increased consumption of both fossil and non-fossil fuels in the long run[22,24]. The dependence of alternative energy's displacement efficiency on its share in the electricity supply mix (Fig. 3a) suggests that the magnitude of the rebound effect is probably between 10 and 50% in China, although more rigorous analysis is needed to validate this estimation. Thus, expanding alternative energy production alone is insufficient and might be counter-productive at reducing the consumption of fossil fuels. It is worth noting that the magnitude of such rebound is within the typical range (20–60%) observed for energy efficiency improvement[14,21,23], and is not high enough to derail alternative energy as a solution for reducing GHG emissions.

Under the pressures of GHG reduction and air pollution mitigation, China has set the goals of raising the share of non-fossil fuels in the national primary energy mix from the current level of ~8 to 15% and reducing that of coal to <62% by 2020 (ref. 29), as well as capping the coal consumption at 4.2 billion tonnes by 2020 (ref. 30). However, with electricity generation consuming over half of the coal mined domestically and the overall weak displacement of fossil-fuel-generated electricity by alternative energy, development of non-fossil energy sources on much more massive scales would be necessary, if the energy use pattern over the past two decades were to continue in the near future. Meanwhile, such expansion would bring some unique and potentially serious risks to the eco-environment and human health[4,11,12], as summarized in Supplementary Table 11. There is also a competition between renewables and nuclear power as clean sources of electricity, while the latter is associated with unique safety concerns and challenges arising from nuclear meltdowns and disposal of the radioactive waste[10]. In particular, the proximity to dense populations, which is typically considered as an advantage of nuclear plants, significantly elevates the risk of radiation release in the events of natural disasters (for example, earthquakes and tsunamis) or terrorist attacks[31]. Clearly, China's future energy need cannot be addressed by simply expanding the alternative energy sources alone. Similarly, increasing the production of non-fossil energy without controlling the consumption of fossil fuels is not expected to reduce GHG emissions in the other developing countries with unmet energy demand, either.

Although debatable, it has been argued that consumption of energy (and other resources as well) within the economy is both a consequence of growth, and a driver of growth as well[22,32]. As indicated by the results of causality tests (Supplementary Table 12), inter-dependence between electricity consumption and economic growth existed in different regions of China.

Taxation and regulation are easy ways to discourage and restrict energy consumption, but such measures would involve economic costs to society and hurt the economic growth[17]. Thus, China should aim for less $CO_2$ emissions, not less energy consumption, while paying greater attention to energy conservation. Improving the efficiency of energy use and promoting energy savings in the industrial, transportation, commercial and residential sectors can help cut the energy demand cost-effectively. Ultimately, energy conservation, that is, prevention of wasteful use, together with the use of renewables, can result in improved environmental quality while sustaining healthy economic growth.

China has managed to achieve a weak decoupling of economic growth from fossil energy consumption and GHG emissions over the past decade, while the approximately stable $CO_2$ emissions and strong economic growth in very recent years indicates that the country has been on the course of strong decoupling[33]. A key driver for the strong decoupling that had occurred in some developed countries (for example, Germany) was the reduced cost of renewables, which accelerated their deployment and facilitated the substitution of fossil fuels[33]. Environmental economists have long argued that efficiency gains must be paired with conservation policies (for example, taxation and cap-and-trade) to reduce consumption by keeping the cost of resource use the same or even higher[34–38]. With 10–50% rebound in the consumption of fossil-fuel-fired electricity from the increased supply of alternative energy observed in China's regional grids, suppressing the rebound's magnitude through proper government interventions can help maximize the benefits from the installation of such energy sources.

Increased production of alternative energy alone is probably less effective at cutting fossil fuel consumption and thus GHG emissions compared to direct suppression, as the former option does not account for the individual and collective behaviours of energy users in the political, social, cultural and economic contexts[11,12,17,19]. Market instruments, including fuel taxes and incentives, can help promote alternative energy while curbing the rebound in consumption of fossil fuels[21]. The most effective strategy for meeting the energy demand in China without increasing GHG emissions is probably a policy integrating carbon taxes, non-fossil fuels and energy efficiency[17,19,21]. The revenue from carbon taxes can subsidize renewable energy for replacing fossil fuels, the cost savings from efficiency gains make government interventions designed to reduce energy demand and promote renewables more affordable and thus more likely to be successfully implemented, while energy consumption and the economy would continue to grow[17,19]. Overall, renewable technologies and energy efficiency, reinforced by effective carbon emissions reduction programs, can accelerate China's transition towards a low-carbon economy with high energy consumption and high efficiency.

## Methods

**Panel data analysis.** We investigated the displacement of fossil-fuel-generated electricity by alternative energy in mainland China using the time series from 30 provinces and municipalities over the period of 1995–2014. We focused specifically on electricity generation because alternative energy development in the country has occurred almost exclusively in this field, with the exception of very limited biofuel production[4]. The displacement efficiency of trans-provincial imported electricity (regardless of the source of generation), which was not considered in York's cross-national longitudinal study, was also analysed, because of the prevalent practice of cross-province transmission in China[2,39,40]. Tibet, in which many areas do not have adequate access to electricity due to the low population density and mountainous terrain, was excluded in the analysis. First-order autocorrelation (Lagram–Multiplier test) was detected in the time series data, while statistically significant impact was found for the dummy variables of entity and time. As a result, two-way fixed-effects panel regression models with Prais–Winsten estimation for first-order autocorrelation were constructed to account for the possible serial correlation as well as the time and entity fixed effects.

Following the pioneering work of York[12], we modelled electricity supply (or consumption) as a linear function of several variables reflecting economic conditions, industrial activities and urbanization. The model for electricity supply is approximated as:

$$F_{it} = \alpha_i + \gamma_t + \beta_{NF}NF_{it} + \beta_{IE}IE_{it} + \beta_{EE}EE_{it} + \beta IF_{it} + u_{it} \qquad (1)$$

where indices $i$ and $t$ denote the individual province and time dimension; $F_{it}$ and $NF_{it}$ are the per capita supply of electricity generated locally (that is, within the provinces) by fossil-fuel and non-fossil-fuel sources, respectively; $IE_{it}$ and $EE_{it}$ are the trans-provincial imported and exported electricity per capita, respectively; $IF_{it}$ is the (unknown) coefficient vector of the regression for the other impact factors (that is, predictor variables), such as GDP per capita, and levels of industrialization and urbanization; $\alpha_i$ and $\gamma_t$ are the intercepts for each province and each year, respectively; $\beta_{IE}$, $\beta_{EE}$ and $\beta$ are the coefficients measuring the impact intensity of the corresponding variables; and $u_{it}$ is the error term, which satisfies:

$$u_{it} = \rho_i u_{i,t-1} + e_{it} \qquad (2)$$

where $e_{it}$ is a white noise (independent and identically distributed normal distribution with mean of zero), and $\rho_i$ is the specific coefficient of first-order autocorrelation for province $i$. The displacement efficiency of alternative energy is indicated by $\beta_{NF}$, and its value is $-1$ in the case of proportional displacement. A $\beta_{NF}$ value of 0 or above indicates no displacement effect, while values between 0 and $-1$ represent partial displacement. $\beta_{IE}$ represents the displacement efficiency of imported electricity, and its value is expected to be within the range of 0 and $-1$ when partial displacement occurs.

The displacement effect of electricity generated by alternative energy was estimated by using fossil-fuel-generated electricity per capita as the dependent variable, and the alternative energy, trans-provincial importation and exportation of electricity, and GDP (adjusted for inflation) per capita as the predictor variables. The estimations were then repeated with the addition of other variables, including the percentage of urban population (variable for level of urbanization) and share of non-agricultural GDP (variable for level of industrialization). Many studies have demonstrated that economic development, industrialization and urbanization are key drivers of energy use[39,41,42]. As expected, strong correlations existed between electricity consumption and the GPD per capita, share of non-agricultural GDP and percentage of urban population in China (Supplementary Fig. 4 and Supplementary Table 13). These factors were initially included to account for the significant variation in the regional economy, as was done in the study by York[12]. Although the entire bundle of predictors worked well in the regression models, models with correlated predictors might not give valid results about any individual correlated predictor, or about which predictors are redundant with respect to the others. Thus the percentage of urban population and share of non-agricultural GDP were dropped out in later models.

**Displacement effect on different scales.** Besides the nationwide analysis, models were also constructed to investigate the displacement efficiency in all of China's six inter-provincial regional power grids. The displacement efficiency of hydropower and that of non-hydro alternative energy were further analysed by splitting the term of $\beta_{NF}NF_{it}$ accordingly. In addition, to validate our model and compare the displacement efficiency of alternative energy in China with the global average over the same time frame, we also estimated the displacement efficiency of alternative energy worldwide over the periods of 1960–2009 and 1995–2013, using the data from the same source as reported in the study of York[12].

**Uncertainties.** With known inadequacies in the regional and national statistical system, some scepticism has been raised about the quality and reliability of China's economic and energy data[43,44]. Nonetheless, the discrepancies and uncertainties in the official statistics are highly unlikely to make a significant difference on the displacement of fossil-fuel-fired generation by alternative energy and trans-provincial electricity transmission, the subject of this study.

**Data availability.** All provincial data of China were obtained from the official sources, including the China Statistical Yearbooks (http://www.stats.gov.cn/tjsj/ndsj/) and the China Energy Statistical Yearbooks (http://tongji.cnki.net/kns55/Navi/NaviDefault.aspx). Data on all countries and regions over the period of 1960–2013 were from the World Development Indicators of the World Bank (http://data.worldbank.org/data-catalog/world-development-indicators). The data that support the findings of this study are also available from the corresponding author upon reasonable request. The analyses were conducted using commands for panel data analysis in STATA statistical software (version 14.0): 'xttest3' for testing the existence of heteroscedasticity, 'xtserial' for testing the existence of serial correlation and 'xtpcse' for building linear cross-sectional time-series models with panel-corrected s.e.'s.

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

## Acknowledgements

This work was supported in parts by the Natural Science Foundation of China (Grant Nos 41322024, 41673089 and 41472324). H.C. was also supported by the National Program for Support of Top-notch Young Professionals and the Changjiang Young Scholar Program.

## Author contributions

Y.H. conceived the study and conducted data collection and analysis. H.C. prepared the main body of the manuscript. Both authors discussed the results and contributed to the editing of the final version of the manuscript.

## Additional information

**Competing financial interests:** The authors declare no competing financial interests.

