## [Peer Review File · Nature Communications]

Reviewers' comments:

Reviewer #1 (Remarks to the Author):

This is a relatively well-executed study on a very important topic. The authors build on York's pioneering work from a few years ago on fossil fuel displacement, and apply the approach to a smaller scale: the province level in China. With moderate revisions it could make a strong contribution to the multidisciplinary research on climate change mitigation. I have some suggestions for revisions.

1. The author's description of York's findings and their implications are not accurate (and to be clear, I am not York).

For example, the authors state: "Instead of offsetting the generation of an equal amount of energy from fossil fuels, much more non-fossil-fuel energy is actually required to achieve the same outcome, due to the complexity of economic system and human behavior. These findings imply that the growth of non-fossil-fuel energy should be much more massive than previously thought to substantially reduce the consumption of fossil fuels."

But York's analysis doesn't necessarily suggest that simply greater growth in non-fossil-fuel energy is the way to reduce consumption of fossil fuels. Rather, significant structural, institutional, and behavioral changes might be what is needed, and not just more growth in renewables.

2. It is unclear if the estimated models are actually two-way fixed effects models, including fixed effects for years (year dummy variables) as well as for the provinces.

3. It is inappropriate to make direct comparisons between the estimates in this study with the estimates from York's cross-national longitudinal study, given the differences in units of analyses, time frames, and included control variables.

4. The sources of the data as well as their known reliability, validity, and quality should be discussed in greater detail for the benefit of readers. Related to this, are these data commonly used in other studies? If so, some citations would be good.

5. The authors should provide a more balanced discussion of nuclear power (e.g., risk).

Reviewer #2 (Remarks to the Author):

The authors aimed to assess the effectiveness of the replacement of alternative energy to fossil fuels in terms of electricity generation. The authors have mentioned some key issues challenges such as the dispatch of renewable electricity, long distance transmission and low efficiency of renewable electricity and etc. However, this article didn't provide a thorough and comprehensive study in any of these topics and draw some in-depth and insightful conclusions. The major conclusions such as "development of alternative energy should focus on local resources and avoid long-distance transmission" and "the displacement efficiency of non-fossil fuel generated and trans-province transported electricity varies largely across China" are more like common sense in this area. They are not well supported by the methodology and in-depth analysis. This article could be improved in several aspects:

1. Dispatch of renewable electricity is a major challenge in Chinese electricity generation system. For example, in Inner Mongolia area (the largest wind base in China), 20-30% of wind electricity is discarded annually which reduced the efficiency of the whole electricity system. The authors have provided some analysis of hydro power and nuclear power yet didn't come across this key issue.

Besides, alternative energy includes a lot of energy sources such as wind, solar, biomass, geothermal and etc. The article only focused on hydro and nuclear.

2. The authors also attempted to analyze trans-province electricity efficiency, which requires a lot of work. In this article, only Zhejiang and Guangdong are analyzed. The authors are suggested to focus the study on different grids of China instead of provinces and cover the whole picture of China's electricity system.

3. Regarding to tables and figures, the authors provided some background information of different provinces such as electricity consumption per capita, GDP per capita, urbanization rate. These background are interesting but not closely link to the contents and analysis of the article. Audience would be more interested in some in-depth information gained from model results such as grid efficiencies of different areas, transmission losses and etc.

Reviewer #3 (Remarks to the Author):

The authors of this paper analyze the extent to which non-fossil-fuel-produced energy displaces fossil-fuel-generated electricity in China using data from 1995-2012. Essentially, the paper is testing the assumption embedded in many policies to promote renewable energy that proportional displacement (one unit of non-fossil-fuel-produced energy replaces one unit of fossil-fuel-produced energy) occurs. Like others before them, they find that perfect displacement (one unit for one unit) does not happen, and that non-fossil-fuel-generated electricity only displaces about 1/3 unit of coal-generated electricity. They also test the extent to which energy transported between provinces displaces energy produced in that province, finding that transported energy displaces only about half of the energy produced in-province.

Overall, this is an interesting analysis done at a relevant scale and in a very important country (China) to the future of GHG emissions. The methods are appropriate and the results confirm (and extend) analyses done in other countries. The paper also provides a valuable contribution by introducing a spatial component and by doing a detailed analysis of a single country that includes a consideration about what type of non-fossil-fuel electricity is generated in different provinces and how that also relates to displacement rates in a variety of ways. However, I think two main large-scale revisions would improve the paper.

First, the authors do not spend very much time discussing or citing the theoretical work on why one-for-one displacement does not occur. There are developed theories about this, such as the Jevon's paradox and debates over "rebound effects" that may either help explain their findings or at least put their study in the context of similar work that has been done in recent years. As of now, the authors make several statements that imply some type of behavioral explanation (for example, they write: "Instead of offsetting the generation of an equal amount of energy from fossil fuels, much more non-fossil-fuel energy is actually required to achieve the same outcome, due to the complexity of economic system and human behavior." (p. 4, lines 78-80) and "Consequently, the electricity generated by such sources was typically not used efficiently for boosting local industrial and socioeconomic development." (p. 5-6, lines 112-114) but do not follow through by providing a specific explanation, or even more detail about what they mean by these statements. I did find a little bit of explanation about the reason trans-province transported energy does not displace locally coal-generated electricity (coal-generated power is higher priority for using because it has high capital costs, and the renewable energy is produced in remote areas), but even here there could be more in way of explanation.

My second suggestion is to reorganize the paper in order to avoid repetitiveness and make it easier to read and follow. The introduction is good, but beginning with the results section, I would recommend beginning with a clear discussion of all of the relevant results, followed by an explanation of why these results matter or are significant and how they might be explained, followed by policy recommendations. Currently, the results section seems to jump between reported results, some brief thoughts about them and potential explanation, and some partial

suggestions for what to do to fix them. As a result, some of the points and/or explanations are repeated in different sections and it is more difficult to identify the paper's main contribution(s).

Smaller questions/suggestions:

One question: Is the trans-province transported energy all non-fossil-fuel-generated, or is this just referring to all energy that is moved between provinces? I wasn't 100% clear on that, so either way it would help to make that more explicit.

This paper could also use a close grammatical edit (though there are only a few grammatical mistakes that I noticed).

Overall, I think the paper is a significant contribution to the very important and timely research being done on energy production/the shift towards non-fossil-fuel-based energy/GHG emissions/climate-related public policy, so I wish the authors the best of luck in preparing a strong revision.

Reviewer #1 (Remarks to the Author):

This is a relatively well-executed study on a very important topic. The authors build on York's pioneering work from a few years ago on fossil fuel displacement, and apply the approach to a smaller scale: the province level in China. With moderate revisions it could make a strong contribution to the multidisciplinary research on climate change mitigation. I have some suggestions for revisions.

We thank the reviewer for the very positive comments on our work! During the revision, we updated our analysis results by incorporating the electricity consumption data of all provinces and municipalities in China (excluding Tibet) in the years of 2013 and 2014, which just became available very recently. We also performed additional model analyses (e.g., global electricity substitution over the periods of 1960-2009 and 1995-2013, electricity substitution in the six regional power grids of China) and discussed the results in light of classical energy economics literature. The significantly expanded and improved results and discussion make this study more solid and compelling.

1. The author's description of York's findings and their implications are not accurate (and to be clear, I am not York). For example, the authors state: "Instead of offsetting the generation of an equal amount of energy from fossil fuels, much more non-fossil-fuel energy is actually required to achieve the same outcome, due to the complexity of economic system and human behavior. These findings imply that the growth of non-fossil-fuel energy should be much more massive than previously thought to substantially reduce the consumption of fossil fuels." But York's analysis doesn't necessarily suggest that simply greater growth in non-fossil-fuel energy is the way to reduce consumption of fossil fuels. Rather, significant structural, institutional, and behavioral changes might be what is needed, and not just more growth in renewables.

We thank the reviewer for pointing this out! Based on the reviewer's comment, we revised the sentences discussing the findings of York's study in the Introduction from:

"A recent study showed that the displacement efficiency of non-fossil-fuel sources and electricity generated from them was less than one-quarter and one-tenth for substituting fossil fuels and fossil-fuel-generated electricity, respectively^{10,11}. Instead of offsetting the generation of an equal amount of energy from fossil fuels, much more non-fossil-fuel energy is actually required to achieve the same outcome, due to the complexity of economic system and

human behavior¹⁰. These findings imply that the growth of non-fossil-fuel energy should be much more massive than previously thought to substantially reduce the consumption of fossil fuels¹¹.”

to:

“A pioneering study by York found that the displacement efficiency of non-fossil-fuel energy sources and the electricity generated from them in 132 nations/regions of the world over the 1960 to 2009 period was less than one-quarter and one-tenth for substituting fossil fuels and fossil-fuel-generated electricity, respectively^{11,12}. Contradictory to the presupposed proportional displacement, more than 11 kWh of non-fossil-fuel electricity was required to displace 1 kWh of fossil-fuel-generated electricity^{11,12}. As the reduction in fossil fuel consumption was far from being commensurate with the increase in alternative energy production, the growth of alternative energy should be much more massive than previously thought to substantially reduce the global consumption of fossil fuels¹¹. These findings also highlight the importance of changing the political, economic, and cultural contexts to facilitate the displacement of fossil fuels and to curb the energy consumption at the individual/household and system levels during the course of renewable energy development^{11,12}.” (Page 5, Lines 97-108)

In addition, we also added the following sentences (with citations to York’s study and other energy economic studies) on the need of changing the political, social, cultural and economic context of energy consumption and policy changes to facilitate China’s transition towards a low carbon economy in the revised manuscript:

“However, with electricity generation consuming over half of the coal mined domestically and the overall weak displacement of fossil-fuel-generated electricity by alternative energy, development of non-fossil energy sources on much more massive scales would be necessary to meet the coal cap, if the energy use pattern over the past two decades were to continue in the near future. Meanwhile, such expansion would bring some unique and potentially serious risks to the eco-environment and human health^{4,11,12}, as summarized in Supplementary Table 11.” (Page 15, Lines 322-328)

“Increased production of alternative energy alone is probably less effective at cutting fossil fuel consumption and thus GHG emissions compared to direct suppression, as the former option does not account for the individual and collective behaviors of energy users in the political, social, cultural, and economic contexts^{11,12,17,19}. Market instruments, including fuel taxes and incentives, can help promote alternative energy while curbing the rebound in consumption of fossil fuels²¹.” (Page 17, Lines 369-374)

“The most effective strategy for meeting the energy demand in China without increasing GHG emissions is probably a policy integrating carbon taxes, non-fossil fuels, and energy efficiency^{17,19,21}. The revenue from carbon taxes can subsidize renewable energy for replacing fossil fuels, the cost savings from efficiency gains make government interventions designed to reduce energy demand and promote renewables more affordable and thus more likely to be successfully implemented, while energy consumption and the economy would continue to grow^{17,19}.” (Page 17, Lines 374-380)

2. It is unclear if the estimated models are actually two-way fixed effects models, including fixed effects for years (year dummy variables) as well as for the provinces.

Yes, two-way fixed effects panel regression models were used in the analysis. To clarify, we revised the sentence describing the panel regression model from:

“We constructed fixed-effects panel regression models with Prais-Winsten correction for first-order autocorrelation to investigate the displacement of fossil-fuel-generated electricity by non-fossil-fuel sources in mainland China, using the time series from 30 provinces and municipalities over the period of 1995-2012.”

to:

“We investigated the displacement of fossil-fuel-generated electricity by alternative energy in mainland China using the time series from 30 provinces and municipalities over the period of 1995-2014.” (Page 18, Lines 388-390)

“First-order autocorrelation (Lagrange-Multiplier test) was detected in the time series data, while statistically significant impact was found for the dummy variables of entity and time. As a result, two-way fixed-effects panel regression models with Prais-Winsten estimation for first-order autocorrelation were constructed to account for the possible serial correlation as well as the time and entity fixed effects.” (Page 18, Lines 397-401)

In addition, we also modified the relevant sentences in the Abstract and Introduction sections to explicitly state the “two-way fixed-effects panel regression models” in the revised manuscript:

“The displacement of fossil-fuel-generated electricity by alternative energy, primarily hydropower, and trans-provincial imported electricity in China between 1995 and 2014 were analyzed using two-way fixed-effects panel regression models.” (Page 1, Lines 23-26)

“In this work, we used two-way fixed-effects models to determine the efficiency at which alternative energy and trans-provincial imported electricity (regardless of the generation source) displacing the electricity generated by local fossil-fuel-fired plants (i.e., within the provinces) both nationwide and in the six regional grids over the period of 1995-2014.” (Page 6, Lines 114-118)

3. It is inappropriate to make direct comparisons between the estimates in this study with the estimates from York's cross-national longitudinal study, given the differences in units of analyses, time frames, and included control variables.

This is a good point! Based on the reviewer's comment, we analyzed the global electricity use data over the same study period and over the periods of 1960-2009 and 1995-2013 using our model to compare the results. The following sentences were added to the revised manuscript:

“For the purpose of comparison, we also investigated the global displacement of fossil-fuel-generated electricity by alternative energy using our model. The displacement coefficient of alternative energy between 1960 and 2009 was found to be -0.136 ± 0.020 , which is close to that (-0.089 ± 0.009) obtained in York's cross-national longitudinal study¹², while that over the 1995 to 2013 period was -0.114 ± 0.018 (Supplementary Table 3). That is, the displacement of fossil-fuel-generated electricity by alternative energy in China was two times more efficient than the global average over the past two decades.” (Page 7, Lines 136-142)

“In addition, to validate our model and compare the displacement efficiency of alternative energy in China with the global average over the same time frame, we also estimated the displacement efficiency of alternative energy worldwide over the periods of 1960-2009 and 1995-2013, using the data from the same source as reported in the study of York¹².” (Page 20, Lines 445-449)

Accordingly, the results from our analysis of the global electricity use data over the periods of 1960-2009 and 1995-2013, in comparison with York's results, are presented in Table 3 of the

Supplementary Material:

Table 3. Estimated parameters for models on the displacement effect of alternative energy on fossil-fuel-generated electricity worldwide over the periods of 1960-2009 and 1995-2013, in comparison with the results obtained in the study of York¹⁰.

Predictor variable	York's study [#]	This study ^{&}	
	1960-2009	1960-2009	1995-2013 [%]
Alternative energy per capita	-0.089* (0.009)	-0.136* (0.020) [<0.001]	-0.114* (0.018) [<0.001]
GDP per capita	296.453* (30.828)	0.139* (0.018) [<0.001]	0.154* (0.017) [<0.001]
(GDP per capita) ²	-8.044* (1.147)	0.000* (0.000) [<0.001]	0.000* (0.000) [<0.001]

[#] The average displacement efficiency of alternative energy on fossil-fuel-generated electricity in a total of 132 countries and regions over the period of 1960-2009 was analyzed by modeling the electricity demand as controlled by GDP per capita in the study of York¹⁰; [&] The present study investigated the displacement efficiency in 133 countries and regions using a similar model and the data from the same source as reported in York's study¹⁰; [%] Global electricity use data in 2014 is not available from the database of World Bank yet at the present time; * Statistically significant at the 0.05 alpha level (two-tailed test); Standard errors are reported in parentheses, while *p*-values are presented in brackets; The cross-sectional and time-series effects are included in all panel models but are not shown here.

In addition, we acknowledged that the displacement efficiency of alternative energy is influenced by many factors by adding the following sentences in the Discussion section of the revised manuscript:

“The results presented above indicate that although far from the proportional displacement desired, alternative energy was utilized two times more efficiently at substituting fossil-fuel-fired electricity generation in China than the global average over the past two decades. This is mainly attributed to the unquenchable thirst for electric power during the course of fast industrialization and urbanization in China, and the trend is expected to continue in the near future. The displacement efficiency of alternative energy can be largely different among countries due to the facts that the growth, technology, and consumption pattern vary significantly, and that the energy consumption of a country evolves as it goes through the stages of economic growth¹⁴. The availability, efficiency, and costs of energy are a significant constraint to industrial activities in the developing countries, particular those with unreliable supply¹⁵. The rapidly rising power demand brought by the strong economic growth and the improving standards of living in China had led to regional and seasonal power shortages in the coastal provinces since the early 2000s⁴. Due to the existence of significant unmet demand, the increased supply of alternative energy would be readily consumed in growing the economy, resulting in relatively high overall efficiency of alternative energy at replacing fossil-fuel-fired generation.” (Pages 10-11, Lines 224-239)

4. The sources of the data as well as their known reliability, validity, and quality should be discussed in greater detail for the benefit of readers. Related to this, are these data commonly used in other studies?

If so, some citations would be good.

We thank the reviewer for pointing this out! We briefly described the data source in the original manuscript with the following sentence:

“...using the time series from 30 provinces and municipalities over the period of 1995-2012.”

“All data were obtained from the official sources, including the China Statistical Yearbooks (economic and population data) and the China Energy Statistical Yearbooks (energy supply and consumption, and energy balance tables).”

Because multiple statistical yearbooks were used, we did not list them one by one in the original manuscript. Following the reviewer’s comment, we listed the links to the databases in the revised manuscript and discussed the reliability, validity, and quality of the data used in this study. Accordingly, the following sub-section was added to the Methods section of the revised manuscript:

“Data availability

All data were obtained from the official sources, including the China Statistical Yearbooks (<http://www.stats.gov.cn/tjsj/ndsj/>) and the China Energy Statistical Yearbooks (<http://tongji.cnki.net/kns55/Navi/NaviDefault.aspx>). With known inadequacies in the regional and national statistical system, some skepticism has been raised about the quality and reliability of China’s economic and energy data^{43,44}. Nonetheless, the discrepancies and uncertainties in the official statistics are highly unlikely to make a significant difference on the displacement of fossil-fuel-fired generation by alternative energy and trans-provincial electricity transmission, the subject of this study.” (*Pages 20-21, Lines 451-460*)

In addition, we also described the data source for the newly added analysis on global electricity use by adding the following sentence:

“In addition, to validate our model and compare the displacement efficiency of alternative energy in China with the global average over the same time frame, we also estimated the displacement efficiency of alternative energy worldwide over the periods of 1960-2009 and 1995-2013, using the data from the same source as reported in the study of York¹²” (*Page 20, Lines 445-449*)

5. The authors should provide a more balanced discussion of nuclear power (e.g., risk).

We thank the reviewer for pointing this out! Based the comments from reviewers, we have thoroughly revised the Discussion section of the revised manuscript. As a result, the following sentences on the advantages of nuclear power had been deleted from the revised manuscript:

“With low variable costs, nuclear power readily replaces coal-fired generation at providing base-load power. Furthermore, nuclear reactors can be built near the demand centers, as long as sufficient cooling water is available, avoiding long-distance transmission.”

Nonetheless, the major advantages and limitations of nuclear power can be found in Table 1 of the Supplementary Material (*Pages S1-S3, not shown here*).

Due to space limitation of the revised manuscript, and the fact that the grid-based analysis results could better illustrate the significant spatial variation in the displacement effect of alternative energy and imported electricity in China, we deleted the case study on displacement effect of hydro and nuclear power in Guangdong and Zhejiang provinces in the revised manuscript.

Following the reviewer's suggestion, we added the following sentences to discuss the risk and challenge of nuclear power in the revised manuscript:

“There is a competition between renewables and nuclear power as clean sources of electricity, while the latter is associated with unique safety concerns and challenges arising from nuclear meltdowns and disposal of the radioactive waste¹⁰. In particular, the proximity to dense populations, which is typically considered as an advantage of nuclear plants, significantly elevates the risk of radiation release in the events of natural disasters (e.g., earthquakes and tsunamis) or terrorist attacks³¹.” (*Page 15, Lines 328-333*)

We also added the following sentence describing the development of nuclear power in China:

“Meanwhile, although falling behind the United States, France, and Japan in generation capacity, nuclear power in China is expanding at the fastest rate, despite of the nuclear power phase-out in some European countries after the Fukushima nuclear disaster in 2011¹⁰.” (*Page 3, Lines 59-62*)

Furthermore, we also compared the advantages and environmental and human health impacts of fossil fuels, nuclear power, and renewable energy sources in a new table (Table 11. Summary of the major advantages and environmental and human health impacts of fossil fuels, nuclear power, and renewable energy sources) in the revised Supplementary Material (*Pages S12-S14, not shown here*).

Finally, in the last section of the revised manuscript (Implications for China's Transition towards a Low-carbon Economy and Policy Recommendations), we also focused the discussion only on renewables for helping China's transition towards a low-carbon economy. Due to space reason, this section is not shown here but can be found on *Pages 14-17, Lines 316-382* of the revised manuscript.

Reviewer #2 (Remarks to the Author):

The authors aimed to assess the effectiveness of the replacement of alternative energy to fossil fuels in terms of electricity generation. The authors have mentioned some key issues challenges such as the dispatch of renewable electricity, long distance transmission and low efficiency of renewable electricity and etc. However, this article didn't provide a thorough and comprehensive study in any of these topics and draw some in-depth and insightful conclusions. The major conclusions such as "development of alternative energy should focus on local resources and avoid long-distance transmission" and "the displacement efficiency of non-fossil fuel generated and trans-province transported electricity varies largely across China" are more like common sense in this area. They are not well supported by the methodology and in-depth analysis. This article could be improved in several aspects:

We appreciate the reviewer's comments. The two major points made by the reviewer are addressed below:

(i) The main purpose of this study:

As mentioned by the reviewer, the main purpose of this study was "to assess the effectiveness of the replacement of alternative energy to fossil fuels in terms of electricity generation". We stated in the original manuscript that:

"In this work, we use two-way fixed effects panel regression modeling to determine (1) the average displacement efficiency of non-fossil-fuel-generated electricity across all the provinces in mainland China (except Tibet) over the past two decades, and (2) the displacement efficiency of trans-province transported electricity. The results provide the very first country-specific estimations on displacement of coal-generated electricity by alternative energy and the use efficiency of trans-province electricity transmission. The findings can help guide the development of electric power and alternative energy sectors on meeting the growing power demand while reducing the dependence on fossil fuels in China."

In the revised manuscript, we revised the above sentences to better state the purpose of this study:

"In this work, we used two-way fixed-effects models to determine the efficiency at which alternative energy and trans-provincial imported electricity (regardless of the generation source) displacing the electricity generated by local fossil-fuel-fired plants (i.e., within the provinces) both nationwide and in the six regional grids over the period of 1995-2014. The results provide the very first estimation on the displacement effect of alternative energy and trans-provincial imported electricity on local fossil-fuel-fired electricity generation in China, a key player on the global energy market. The findings also reveal important insights on effective policies to facilitate decoupling of economic growth from fossil energy consumption and thus GHG emissions in China." (Page 6, Lines 114-122)

The reviewer commented that: "The authors have mentioned some key issues challenges such as the dispatch of renewable electricity, long distance transmission and low efficiency of renewable electricity and etc. However, this article didn't provide a thorough and comprehensive study in any of these topics and draw some in-depth and insightful conclusions." We briefly discussed "dispatch of renewable electricity", "long distance transmission", and "low efficiency of renewable electricity" as possible causes for the low displacement efficiency of alternative energy and low efficiency for trans-province electricity

transmission in the following sentences of the revised manuscript:

“The electric power industry has been developed from and is well accustomed to fossil-fuel-fired power generation, while the infrastructure, production cost, and supply characteristics of alternative energy sources can be significantly different. Based on their costs of generation and the flexibility in meeting the peak load, nuclear power and hydropower are often adopted as baseload resources, followed by coal, natural gas, and oil-fired generation (Supplementary Table 1). In contrast, although wind energy and solar power typically have low variable costs, they are inherently weather-dependent and intermittent, and their supply does not follow the typical demand curve, which makes their integration into the power grid challenging⁴. Furthermore, large renewable energy sources (e.g., hydro and wind) are mostly located in the remote areas, and long-distance transmission networks are often required to transmit the electricity to the load centers in major cities and industrial hubs, where the electricity demand is often met by local coal plants. Approximately 6-7% of the electricity fed into the grid was lost in transmission and distribution in China (Supplementary Fig. 1), while renewable electricity generated in some remote regions was not grid-connected due to insufficient transmission capacity. It has been reported that the northwestern provinces and regions rich in wind power resources, including Inner Mongolia, accounted for three-quarters of China’s unconnected wind power generation⁴. In addition, when sourcing power, state-owned electrical grid companies often gave priority to the electricity generated by local coal plants, which had relatively high capital costs, over the electricity supplied from renewable sources or imported from the other provinces. Therefore, besides technological limitations, a range of practical, economic, political factors, and the subjective considerations of the grid operators, can all influence the actual substitution of fossil-fuel-generated electricity by alternative energy.” (Pages 4-5, Lines 73-95)

“A thorough and comprehensive study” on the topics of “*dispatch of renewable electricity, long distance transmission and low efficiency of renewable electricity*” was not the purpose of this study. Instead, the primarily objective of this study was to quantify the displacement efficiency of electricity generated from alternative energy sources on fossil-fuel-generated electricity in China. The relatively low displacement efficiency observed is mainly attributed to the rebound in the consumption of fossil-fuel-generated electricity with the exogenous supply of alternative energy. To clarify, we added the following paragraph in the revised manuscript that:

“In addition to factors such as transmission and distribution losses and under-utilization, the inefficient substitution of fossil-fuel-generated electricity by alternative energy could also be contributed by the consumption behaviors of those that demand energy. The Jevons Paradox, that is, increasing the productivity of a commodity leads to greater consumption, has long been documented in energy economics and policy research^{14,17,18}. The energy savings achieved from improved energy efficiency can be partially or even fully cancelled out due to greater consumption of energy stimulated by the lower cost of energy services¹⁸⁻²⁰. A substantial amount of empirical evidence supports the existence of such rebound effect, which can be considered as a behavioral or other systemic response to improved energy efficiency^{14,15,18,20-22}. Nonetheless, the magnitude of the rebound effect is limited, typically offsetting 20-60% of the expected energy savings^{14,21,23}. Although very little research has investigated the relationship between renewable energy generation and energy consumption, an exogenous increase in the supply of energy from non-fossil-fuel sources, can also induce greater energy consumption, i.e., rebound effect, in a way similar to that of energy efficiency^{24,25}. Studies have suggested that the ethanol

subsidy in the United States could lower the price of E10 (10% ethanol and 90% gasoline), which might eventually result in greater gasoline demand and more oil imports^{26,27}. Rebound effect was also observed for residential electricity consumption in the states adopting one or more financial incentives for renewable energy in the United States²⁸.”
(Pages 13-14, Lines 279-296)

(ii) The main conclusion of this study:

The reviewer commented that: “*The major conclusions such as “development of alternative energy should focus on local resources and avoid long-distance transmission” and “the displacement efficiency of non-fossil fuel generated and trans-province transported electricity varies largely across China” are more like common sense in this area. They are not well supported by the methodology and in-depth analysis*”. Indeed, the statements of “development of alternative energy should focus on local resources and avoid long-distance transmission” and “the displacement efficiency of non-fossil fuel generated and trans-province transported electricity varies largely across China” were based on the results obtained in this study. These statements were based on the results obtained through our model analysis and are well supported by the findings of this study, as detailed below:

- (a) “*development of alternative energy should focus on local resources and avoid long-distance transmission*” is based on the results that show trans-province electricity transmission has relatively low efficiency (Tables 1-3), thus utilization of local energy resources could improve the overall efficiency of energy use.
- (b) “*the displacement efficiency of non-fossil fuel generated and trans-province transported electricity varies largely across China*” is based on comparison of the displacement efficiency in the six regional power grid and in the provinces ranked among the top and bottom halves in hydropower production per capita (Tables 2-4), which show the significant variation in the use efficiency of alternative energy in different regions/provinces across China.
- (c) The fact that the findings are “*more like common sense in this area*” serves to validate the quantitative results obtained from our model analysis. On the other hand, these were not the MAJOR conclusion of this study, even in the original submission.
- (d) To clarify, we stated in the revised manuscript that:
“The above results indicate that exogenous electricity (either generated by alternative energy sources locally or imported from other provinces) could be utilized more efficiently in regions with significant unmet demand. This calls for significant improvement on energy efficiency in the provinces with surplus generation and expansion of the long-distance transmission networks to re-route the surplus electricity more efficiently across the regional grids.” (Pages 12-13, Lines 272-277)

Furthermore, based on the comments from the three reviewers, we have thoroughly re-organized the Results section and the Discussion section in the revised manuscript. Due to space reason, the revised version of these two sections are not shown here, but can be found on Pages 6-14, Lines 124-314 of the revised manuscript.

1. Dispatch of renewable electricity is a major challenge in Chinese electricity generation system. For example, in Inner Mongolia area (the largest wind base in China), 20-30% of wind electricity is discarded annually which reduced the efficiency of the whole electricity system. The authors have provided some analysis of hydro power and nuclear power yet didn't come across this key issue. Besides, alternative energy includes a lot of energy sources such as wind, solar, biomass, geothermal

and etc. The article only focused on hydro and nuclear.

We briefly discussed the dispatch and long-distance transmission issues for renewables in the original manuscript:

“In contrast, although wind energy and solar power typically have low variable costs, they are inherently weather-dependent and intermittent, and their supply does not follow the typical demand curve, which makes their integration into the power grid challenging⁴. Furthermore, large renewable energy sources (e.g., hydro and wind) are mostly located in the remote areas, and long-distance transmission networks are often required to transmit the electricity to the load centers in major cities and industrial hubs, where the electricity demand is often met by local coal plants.” (Page 4, Lines 78-84)

Based on the reviewer’s comment, we added the following discussion on the transmission and distribution losses in China and on the discarded wind power due to lack of grid connection in the revised manuscript:

“Approximately 6-7% of the electricity fed into the grid was lost in transmission and distribution in China (Supplementary Fig. 1), while renewable electricity generated in some remote regions was not grid-connected due to insufficient transmission capacity. It has been reported that the northwestern provinces and regions rich in wind power resources, including Inner Mongolia, accounted for three-quarters of China’s unconnected wind power generation⁴.” (Pages 4-5, Lines 84-89)

Accordingly, we showed the transmission and distribution losses of electricity in China in Figure 1 of the revised Supplementary Material (Page S17):

Figure 1. Average electricity transmission and distribution losses in China between 2000 and 2014. China’s average electricity transmission and distribution losses in recent years were close to those of the United States (about 6%). Data are from electricity balance table on the website of National Bureau of Statistics (<http://data.stats.gov.cn/easyquery.htm?cn=C01>).

It should be noted that renewable energy that was not put into the electric grid, which is the case of the wasted wind energy, did not contribute to the transmission and distribution losses, because it

was not even counted.

This study only analyzed the overall displacement efficiency of alternative energy, and that of hydropower. The displacement efficiency of other alternative energy sources (e.g., wind energy and solar power) could not be analyzed because they gained significant development only recently. Thus valid regression models could not be built with the limited time series and cross-sectional data for the individual non-hydro alternative energy sources. We stated in the original manuscript that:

“Hydro and nuclear power accounted for 17.5 and 1.95% of China’s electricity production, respectively, while the contribution from solar, wind, biomass, and geothermal combined merely reached 2.5% by the end of 2012¹.”

During the manuscript revision, we updated the time series data to the year of 2014, which just became available very recently. Thus, the above sentence was also updated as:

“Hydropower and nuclear power accounted for 18.8 and 2.3% of China’s electricity production, respectively, while the contribution from solar, wind, biomass, and geothermal combined merely reached 2.8% by the end of 2014².” (*Page 3, Lines 53-55*)

We also stated in the revised manuscript that:

“Accounting for over three-quarter of the electricity generation from non-fossil-fuel sources in China, the displacement effect of hydropower is of particular importance.” (*Page 8, Lines 167-168*)

Due to the very small shares of wind power, solar energy, geothermal power, and biomass energy in China’s electricity production over the period of 1995 and 2014, they were lumped together as non-hydro alternative energy sources and discussed together. Their displacement effect on fossil-fuel-generated electricity was discussed in the following paragraph of the original manuscript:

“Models 4-6 indicate they did not have statistically significant displacement effect on fossil-fuel-generated electricity. This is consistent with the fact that large scale deployment of non-hydro alternative energy sources barely occurred over the period of 1995 and 2012 (Supplementary Figure 1). Furthermore, some non-hydro alternative energy sources, such as solar energy and wind power, are inherently weather-dependent and variable, and their supply usually does not follow the typical demand curve³. Thus, they cannot readily substitute coal-fired plants at supplying base-load, and integrating them into the power grid requires adequate, flexible sources of generation to smooth out the variations in their output. Despite of a few pumped hydroelectric storage facilities, the overall lack of energy storage in China has limited the penetration of variable renewable energy sources in the electric grid³.” (These sentences had been deleted or thoroughly changed in the revised manuscript)

In the revised manuscript, the displacement effect of non-hydro alternative energy sources is presented in Tables 3 and 4, and discussed in the following sentences:

“Table 3 shows that the national average displacement coefficient of hydropower (-0.637 ± 0.091) is almost three times that of alternative energy (-0.231 ± 0.078), while no displacement effect is observed for non-hydro alternative energy (see Supplementary Table 7 for full results). The lack of significant displacement effect by non-hydro alternative energy is consistent with the fact that large scale deployment of such energy sources barely occurred in China over the period of 1995-2014 (Supplementary Fig. 2), which is also consistent with the global alternative energy substitution pattern observed in York’s study¹².” (*Page 8, Lines*

168-175)

“Non-hydro alternative energy did not show statistically significant displacement effect in any of the six regional grids, in which its mean share was at most 6.6% (Supplementary Table 8).” (Page 9, Lines 186-187)

“In contrast, no general trend could be found for non-hydro alternative energy, which had no statistically significant displacement effect on fossil-fuel-generated electricity.” (Page 10, Lines 211-213)

“Similarly, with low mean shares in the supply mixes, non-hydro alternative energy sources did not show significant displacement effect on fossil-fuel-generated electricity in any of the regional grids.” (Page 12, Lines 250-252)

2. *The authors also attempted to analyze trans-province electricity efficiency, which requires a lot of work. In this article, only Zhejiang and Guangdong are analyzed. The authors are suggested to focus the study on different grids of China instead of provinces and cover the whole picture of China's electricity system.*

We appreciate the reviewer's comment! Unfortunately, there appears to be some misunderstanding. Throughout the original manuscript, we investigated the overall displacement efficiency of alternative energy on fossil-fuel-fired power generation nationwide (except Tibet), and the overall efficiency of trans-provincial electricity transmission on replacing local fossil-fuel-fired power generation. That is, this study covered the electricity system and trans-provincial electricity transmission in all provinces and municipalities of China, except Tibet. We stated in the original manuscript that:

“We investigated the displacement of fossil-fuel-generated electricity by alternative energy in mainland China using the time series from 30 provinces and municipalities over the period of 1995-2014.” (Page 18, Lines 388-390)

We stated the reason for excluding Tibet in the original manuscript:

“Tibet, in which many areas do not have adequate access to electricity due to the low population density and mountainous terrain, was excluded in the analysis.” (Page 18, Lines 395-397)

The investigation on the displacement efficiency of hydro and nuclear power in Zhejiang and Guangdong is just a case study. Due to space limitation of the revised manuscript, and the fact that the grid-based analysis results could better illustrate the significant spatial variation in the displacement effect of alternative energy and imported electricity in China, we deleted the case study on displacement effect of hydro and nuclear power in Guangdong and Zhejiang provinces in the revised manuscript.

Following the reviewer's suggestion, we also investigated the use efficiency of electricity in all six inter-provincial regional power grids of China. The grid-based results are presented in Tables 2 and 3, and Figure 2 of the revised manuscript:

Table 2. Estimated displacement coefficients of alternative energy and trans-provincial imported electricity in China's six inter-provincial regional power grids.

Predictor variable	Inter-provincial regional power grid					
	East	Central	North	Northeast	Northwest	South
Alternative energy per capita	-0.709* (0.142) [<0.001]	-0.672* (0.065) [<0.001]	3.638* (1.267) [0.004]	3.960* (0.608) [<0.001]	-0.488* (0.099) [<0.001]	-0.888* (0.072) [<0.001]
Trans-provincial imported electricity per capita	-0.197* (0.078) [0.012]	-0.457* (0.105) [<0.001]	-0.841* (0.083) [<0.001]	-0.339 (0.411) [0.409]	0.816* (0.311) [0.009]	-0.891* (0.108) [<0.001]

Results are based on panel analyses of data from 30 provinces and municipalities during 1995-2014, and the electricity demand is modeled as controlled by GDP per capita; * Statistically significant at the 0.05 alpha level (two-tailed test); Standard errors are reported in parentheses, while p -values are presented in brackets.

Table 3. Estimated displacement coefficients of hydropower and non-hydro alternative energy sources nationwide and in the six inter-provincial regional power grids, along with those of trans-provincial imported electricity.

Predictor variable	Nationwide	Inter-provincial regional power grid					
		East	Central	North	Northeast	Northwest	South
Hydropower per capita	-0.637* (0.091) [<0.001]	-1.168* (0.117) [<0.001]	-0.682* (0.064) [<0.001]	5.937* (1.909) [0.002]	1.160 (1.008) [0.250]	-0.543* (0.097) [<0.001]	-0.896* (0.073) [<0.001]
Non-hydro alternative energy per capita	2.399* (0.657) [<0.001]	0.064 (0.206) [0.756]	-1.251 (0.886) [0.158]	3.167* (1.358) [0.020]	4.447* (0.642) [<0.001]	1.911* (0.824) [0.020]	-0.615 (0.575) [0.285]
Trans-provincial imported electricity per capita [§]	-0.271* (0.106) [0.011]	-0.180* (0.075) [0.016]	-0.464* (0.105) [<0.001]	-0.772* (0.087) [<0.001]	-0.170 (0.347) [0.625]	0.532 (0.338) [0.116]	-0.886* (0.109) [<0.001]

Results are based on panel analyses of data from 30 provinces and municipalities during 1995-2014, and the electricity demand is modeled as controlled by GDP per capita; [§] As expected, the displacement coefficients for trans-provincial imported electricity nationwide and in the six regional grids are comparable to those obtained by the previous models (Tables 1 and 2); * Statistically significant at the 0.05 alpha level (two-tailed test); Standard errors are reported in parentheses, while p -values are presented in brackets.

(a)

(b)

(c)

Figure 2. Displacement effect of alternative energy as a function of its penetration in the electricity supply mix of China's six inter-provincial regional power grids: (a) alternative energy; (b) hydropower; and (c) non-hydro alternative energy. Statistically significant and insignificant data are represented by the symbols of “◇” and “Δ”, respectively.

Accordingly, we discussed the results on the displacement efficiency of alternative energy, hydropower, and non-hydro alternative energy, as well as that of trans-provincial imported electricity by adding following paragraphs to the revised manuscript:

“Instead of a unified national grid system, China has six wide area synchronous grids, although the under-developed long-distance transmission capacity has prevented efficient re-routing of electricity from the regions with surplus generation to those in shortages⁴. Table 2 compares the displacement coefficients of alternative energy and trans-provincial imported electricity in the six inter-provincial regional power grids (see Supplementary Table 4 for full results). With the exception of the north and northeast grids, which had low shares (<8%) of alternative energy in their supply mixes (Supplementary Table 5), relatively good to strong displacement effect was observed for alternative energy in the others (-0.488 to -0.888). The imported electricity showed no or statistically insignificant effect on replacing local fossil-fuel-fired power generation in the northwest and northeast grids, which had relatively high exportation of electricity (Supplementary Table 6). Strong displacement occurred in the south grid (-0.891 ± 0.108) and north grid (-0.841 ± 0.083), while the central grid (-0.457 ± 0.105) and east grid (-0.197 ± 0.078) exhibited much poorer displacement. Together, the above results indicate that strong displacement could be achieved by both alternative energy and inter-provincial transmission in some grids, while the national average levels were significantly reduced by the poor efficiency in the rest.” (Pages 7-8, Lines 148-163)

“As expected, the displacement coefficient of hydropower exhibited very large variation among the regional grids. Essentially proportional displacement occurred in the east grid (-1.168 ± 0.117), and strong displacement took place in the south grid (-0.896 ± 0.073), followed by the central grid (-0.682 ± 0.064) and northwest grid (-0.543 ± 0.097). In contrast, no statistically significant displacement was observed in the north and northeast grids, which had rather low shares (<5%) of hydropower in their electricity supply mixes (Supplementary Table 8). As shown in Fig. 1b, the distribution of hydropower resources in China is highly uneven, and so is the hydropower generation capacity¹³. It appears that hydropower had little displacement effect on fossil-fuel-fired generation in the provinces with low production. Non-hydro alternative energy did not show statistically significant displacement effect in any of the six regional grids, in which its mean share was at most 6.6% (Supplementary Table 8). ” (Pages 8-9, Lines 177-187)

“The displacement efficiency of fossil-fuel-generated electricity by alternative energy and trans-provincial imported electricity varied largely across China's six regional grids. Fig. 2a depicts the relationship between the displacement coefficient of alternative energy and its share in the grid's electricity supply mix. It is obvious that alternative energy had no displacement effect on fossil-fuel-generated electricity when its share was very low, while relatively good to strong displacement (-0.488 to -0.888) was observed once it exceeded ~10%.” (Pages 9-10, Lines 201-206)

“Figs. 2b and 2c show the relationship between the displacement coefficients of hydropower and non-hydro alternative energy and their shares in the grid's electricity supply mix, respectively. As expected, the displacement effect of hydropower resembles that of

alternative energy because of its heavy dominance in alternative energy. In contrast, no general trend could be found for non-hydro alternative energy, which had no statistically significant displacement effect on fossil-fuel-generated electricity. Similarly, the displacement coefficient of trans-provincial imported electricity had no apparent dependence on the share of imported electricity or that of net inflow in the grid's electricity supply mix (Supplementary Figs. 3a and 3b). These results consistently suggest that significant penetration of alternative energy on the grid might be necessary for it to achieve meaningful displacement of fossil-fuel-generated electricity. For imported electricity, its use efficiency depended primarily on the region, and good displacement could occur in both regions with net inflow of electricity (e.g., north grid) and those with net outflow of electricity (e.g., central grid)." (Page 10, Lines 208-220)

Due to space limitation of the revised manuscript, and the fact that the grid-based analysis results could better illustrate the significant spatial variation in the displacement effect of alternative energy and imported electricity in China, we deleted the case study on displacement effect of hydro and nuclear power in Guangdong and Zhejiang provinces in the revised manuscript.

3. Regarding to tables and figures, the authors provided some background information of different provinces such as electricity consumption per capita, GDP per capita, urbanization rate. These background are interesting but not closely link to the contents and analysis of the article. Audience would be more interested in some in-depth information gained from model results such as grid efficiencies of different areas, transmission losses and etc.

The original manuscript had 3 tables and 2 figures:

- (i) All 3 tables summarize the results of our model estimation and are not background information.
- (ii) Figure 1 shows the spatial variation in electricity supply and consumption, hydropower production, and GDP per capita in mainland China (excluding Tibet). Although this information can be considered as background, it illustrates several key points: (a) the electricity generation and consumption in China are spatially unbalanced and long-distance electricity transmission is a common practice; (b) the economic development in China is also highly unbalanced; and (c) hydropower generation is concentrated in certain parts of China. All the above information is useful for interpretation of our model results.
- (iii) The reviewer is probably referring to Figure 2 (Scatter plot matrix depicting the correlations between electricity consumption per capita, GDP per capita, share of non-agricultural GDP, and percentage of urban population across 30 provinces and municipalities in China). This figure illustrates the existence of correlations among the predictor variables used in the model analyses. Following the reviewer's comment, we moved Figure 2 to the revised Supplementary Material (as Figure 4, Page S20).

Based on the reviewer's suggestion, we also made the following changes in the revised manuscript:

- (i) We added the following discussion on the transmission and distribution losses in China:
"Approximately 6-7% of the electricity fed into the grid was lost in transmission and distribution in China (Supplementary Fig. 1), while renewable electricity generated in some remote regions was not grid-connected due to insufficient transmission capacity." (Page 4, Lines 84-87)

Accordingly, we showed the transmission and distribution losses of electricity in China in

Figure 1 of the revised Supplementary Material (Page S17):

Figure 1. Average electricity transmission and distribution losses in China between 2000 and 2014. China's average electricity transmission and distribution losses in recent years were close to those of the United States (about 6%). Data are from electricity balance table on the website of National Bureau of Statistics (<http://data.stats.gov.cn/easyquery.htm?cn=C01>).

- (ii) We investigated the displacement efficiency of alternative energy and imported electricity in all the six inter-provincial regional power grids of China (Tables 2 and 3, and Figure 2 of the revised manuscript, and the corresponding discussions added to the revised manuscript). Details of these new results and discussions are presented under our response to Comment #2 from the Reviewer and are not repeated here.
- (iii) We also redrew Figure 1 as two sub-figures, depicting the distribution of electricity generation and consumption, and GDP per capita in China on Fig. 1a, and distribution of hydropower resources and major hydropower stations in China, along with the boundaries of the six inter-provincial regional power grids on Fig. 1b. Hydropower was emphasized separately because it accounted for more than three-quarter of alternative energy production in China. The updated Figure 1 is shown below:

(a)

(b)

Figure 1. Spatial variation in electricity supply and consumption, and hydropower production in China: (a) distribution of electricity generation and consumption, and GDP per capita in mainland China (excluding Tibet) in the year of 2014; and (b) distribution of technically exploitable hydropower resources, hydropower generation (2014), and major hydropower stations in China, along with the boundaries of the six inter-provincial regional power grids (although Tibet is part of the northwest grid, it is excluded in this study due to lack of sufficient data). Around one-third of the coal electricity in China is generated by the power plants located right near large coal mines in the northern and western provinces⁴, whereas over 40% of the electricity is consumed by the major industrial and urban centers in the coastal provinces, which account for about half of China's GDP and industrial output. Most of the

hydropower bases are located in the central and southern provinces, including Sichuan and Yunnan, while the northeastern and northern provinces have very low hydropower generation.

The following sentences in the revised manuscript refer to the information present in the revised Figure 1:

“In addition, no previous research has investigated the use efficiency of trans-provincial transported electricity, which plays a vital role in balancing the significant spatial mismatch of power generation and consumption in the country (Fig. 1a).” (*Pages 5-6, Lines 111-114*)

“As shown in Fig. 1b, the distribution of hydropower resources in China is highly uneven, and so is the hydropower generation capacity¹³.” (*Page 9, Lines 183-184*)

Reviewer #3 (Remarks to the Author):

The authors of this paper analyze the extent to which non-fossil-fuel-produced energy displaces fossil-fuel-generated electricity in China using data from 1995-2012. Essentially, the paper is testing the assumption embedded in many policies to promote renewable energy that proportional displacement (one unit of non-fossil-fuel-produced energy replaces one unit of fossil-fuel-produced energy) occurs. Like others before them, they find that perfect displacement (one unit for one unit) does not happen, and that non-fossil-fuel-generated electricity only displaces about 1/3 unit of coal-generated electricity. They also test the extent to which energy transported between provinces displaces energy produced in that province, finding that transported energy displaces only about half of the energy produced in-province.

Overall, this is an interesting analysis done at a relevant scale and in a very important country (China) to the future of GHG emissions. The methods are appropriate and the results confirm (and extend) analyses done in other countries. The paper also provides a valuable contribution by introducing a spatial component and by doing a detailed analysis of a single country that includes a consideration about what type of non-fossil-fuel electricity is generated in different provinces and how that also relates to displacement rates in a variety of ways. However, I think two main large-scale revisions would improve the paper.

We thank the reviewer for the very positive comments on our work! During the revision, we updated our analysis results by incorporating the electricity consumption data of all provinces and municipalities in China (excluding Tibet) in the years of 2013 and 2014, which just became available very recently. We also performed additional model analyses (e.g., global electricity substitution over the periods of 1960-2009 and 1995-2013, electricity substitution in the six regional power grids of China) and discussed the results in light of classical energy economics literature. The significantly expanded and improved results and discussion make this study more solid and compelling.

1. First, the authors do not spend very much time discussing or citing the theoretical work on why one-for-one displacement does not occur. There are developed theories about this, such as the Jevon's paradox and debates over "rebound effects" that may either help explain their findings or at least put their study in the context of similar work that has been done in recent years. As of now, the authors make several statements that imply some type of behavioral explanation (for example, they write: "Instead of offsetting the generation of an equal amount of energy from fossil fuels, much more non-fossil-fuel energy is actually required to achieve the same outcome, due to the complexity of economic system and human behavior." (p. 4, lines 78-80) and "Consequently, the electricity generated by such sources was typically not used efficiently for boosting local industrial and socioeconomic development." (p. 5-6, lines 112-114) but do not follow through by providing a specific explanation, or even more detail about what they mean by these statements. I did find a little bit of explanation about the reason trans-province transported energy does not displace locally coal-generated electricity (coal-generated power is higher priority for using because it has high capital costs, and the renewable energy is produced in remote areas), but even here there could be more in way of explanation.

We are grateful to the reviewer's helpful suggestion and comments! Following the reviewer's comment, we have thoroughly revised our discussion by incorporating the more fundamental energy economics literatures. The following paragraphs were added to the Discussion section of the revised manuscript:

“In addition to factors such as transmission and distribution losses and under-utilization, the inefficient substitution of fossil-fuel-generated electricity by alternative energy could also be contributed by the consumption behaviors of those that demand energy. The Jevons Paradox, that is, increasing the productivity of a commodity leads to greater consumption, has long been documented in energy economics and policy research^{14,17,18}. The energy savings achieved from improved energy efficiency can be partially or even fully cancelled out due to greater consumption of energy stimulated by the lower cost of energy services¹⁸⁻²⁰. A substantial amount of empirical evidence supports the existence of such rebound effect, which can be considered as a behavioral or other systemic response to improved energy efficiency^{14,15,18,20-22}. Nonetheless, the magnitude of the rebound effect is limited, typically offsetting 20-60% of the expected energy savings^{14,21,23}. Although very little research has investigated the relationship between renewable energy generation and energy consumption, an exogenous increase in the supply of energy from non-fossil-fuel sources, can also induce greater energy consumption, i.e., rebound effect, in a way similar to that of energy efficiency^{24,25}. Studies have suggested that the ethanol subsidy in the United States could lower the price of E10 (10% ethanol and 90% gasoline), which might eventually result in greater gasoline demand and more oil imports^{26,27}. Rebound effect was also observed for residential electricity consumption in the states adopting one or more financial incentives for renewable energy in the United States²⁸.

Similar to the case of efficiency improvement, with technological advances in alternative energy generation and expansion of the production capacity, the costs of energy services (with or without subsidies) are reduced, which can result in increased consumption of both fossil and non-fossil fuels in the long run^{22,24}. The dependence of alternative energy’s displacement efficiency on its share in the electricity supply mix (Fig. 2a) suggests that the magnitude of the rebound effect was probably between 10 to 50% in China, although more rigorous analysis is needed to validate this estimation. That is, alternative energy had little displacement effect on fossil-fuel-generated electricity during the early stage of development. Once it made up more than ~10% of the electricity supply on the grid, alternative energy could have good displacement effect on fossil-fuel-generated electricity, probably due to the scales of economies and the learning effects^{22,24}. Nonetheless, proportional displacement was compromised by the occurrence of the rebound effect, which cancelled out 10-50% of the technologically achievable reduction in fossil-fuel-fired generation. Thus, expanding alternative energy production alone is insufficient and might be counterproductive at reducing the consumption of fossil fuels. It is worth noting that the magnitude of such rebound is within the typical range (20-60%) observed for energy efficiency improvement^{14,21,23}, and is not high enough to derail alternative energy as a solution for reducing GHG emissions.”
(Pages 13-14, Lines 279-314)

In addition, we also added a new section discussing the implications of the results of this study on China’s energy policies in the revised manuscript:

“Implications for China’s Transition towards a Low-carbon Economy and Policy Recommendations

Under the pressures of GHG reduction and air pollution mitigation, China has set the goals of raising the share of non-fossil fuels in the national primary energy mix from the current level of ~8% to 15% and reducing that of coal to <62% by 2020²⁹, as well as capping the coal consumption at 4.2 billion tonnes by 2020³⁰. However, with electricity generation consuming over half of the coal mined domestically and the overall weak displacement of fossil-fuel-generated electricity by alternative energy, development of non-fossil energy

sources on much more massive scales would be necessary to meet the coal cap, if the energy use pattern over the past two decades were to continue in the near future. Meanwhile, such expansion would bring some unique and potentially serious risks to the eco-environment and human health^{4,11,12}, as summarized in Supplementary Table 11. There is a competition between renewables and nuclear power as clean sources of electricity, while the latter is associated with unique safety concerns and challenges arising from nuclear meltdowns and disposal of the radioactive waste¹⁰. In particular, the proximity to dense populations, which is typically considered as an advantage of nuclear plants, significantly elevates the risk of radiation release in the events of natural disasters (e.g., earthquakes and tsunamis) or terrorist attacks³¹. Clearly, China's future energy need cannot be addressed by simply expanding the alternative energy sources alone. Similarly, increasing the production of non-fossil energy without controlling the consumption of fossil fuels is not expected to reduce GHG emissions in the other developing countries with unmet demand, either.

Although debatable, it has been argued that consumption of energy (and other resources as well) within the economy is both a consequence of growth, and a driver of growth as well^{22,32}. As indicated by the results of causality tests (Supplementary Table 12), inter-dependence between electricity consumption and economic growth existed in different regions of China. Taxation and regulation are easy ways to discourage and restrict energy consumption, but such measures would involve economic costs to society and hurt the economic growth¹⁷. Thus, China should aim for less CO₂ emissions, not less energy consumption, while paying greater attention to energy conservation. Improving the efficiency of energy use and promoting energy savings in the industrial, transportation, commercial, and residential sectors can help cut the energy demand cost-effectively. Ultimately, energy conservation, i.e., prevention of wasteful use, together with the use of renewables, can result in improved environmental quality while sustaining healthy economic growth.

China has managed to achieve a weak decoupling of economic growth from fossil energy consumption and GHG emissions over the past decade, while the approximately stable CO₂ emissions and strong economic growth in very recent years indicates that the country has been on the course of strong decoupling³³. A key driver for the strong decoupling that had occurred in some developed countries (e.g., Germany) was the reduced cost of renewables, which accelerated their deployment and facilitated the substitution of fossil fuels³³. Environmental economists have long contended that improved efficiency, instead of lowering resource consumption, may lead to higher production and consumption (i.e., the Jevons paradox), and proposed that efficiency gains must be paired with conservation policies (e.g., taxation and cap-and-trade), which can reduce consumption by keeping the cost of resource use the same or even higher³⁴⁻³⁸. Similarly, rebound in the consumption of fossil fuels could also occur along with the increased supply of renewables (and nuclear power) at lower costs, partially offsetting the displacement. With 10-50% rebound in the consumption of fossil-fuel-fired electricity from the increased supply of alternative energy, suppressing the rebound's magnitude through proper government interventions can reap greater benefits from the installation of such energy sources.

Increased production of alternative energy alone is probably less effective at cutting fossil fuel consumption and thus GHG emissions compared to direct suppression, as the former option does not account for the individual and collective behaviors of energy users in the political, social, cultural, and economic contexts^{11,12,17,19}. Market instruments, including fuel taxes and incentives, can help promote alternative energy while curbing the rebound in consumption of

fossil fuels²¹. The most effective strategy for meeting the energy demand in China without increasing GHG emissions is probably a policy integrating carbon taxes, non-fossil fuels, and energy efficiency^{17,19,21}. The revenue from carbon taxes can subsidize renewable energy for replacing fossil fuels, the cost savings from efficiency gains make government interventions designed to reduce energy demand and promote renewables more affordable and thus more likely to be successfully implemented, while energy consumption and the economy would continue to grow^{17,19}. Overall, renewable technologies and energy efficiency, reinforced by carbon emissions reduction programs, can accelerate China's transition towards a low-carbon economy with high energy consumption and high efficiency." (Pages 14-17, Lines 316-382)

2. *My second suggestion is to reorganize the paper in order to avoid repetitiveness and make it easier to read and follow. The introduction is good, but beginning with the results section, I would recommend beginning with a clear discussion of all of the relevant results, followed by an explanation of why these results matter or are significant and how they might be explained, followed by policy recommendations. Currently, the results section seems to jump between reported results, some brief thoughts about them and potential explanation, and some partial suggestions for what to do to fix them. As a result, some of the points and/or explanations are repeated in different sections and it is more difficult to identify the paper's main contribution(s).*

We greatly appreciate the reviewer's suggestion! In the revised manuscript, we have incorporated results of additional analysis (displacement effect in the six trans-provincial regional power grids, dependence of the displacement effect of alternative energy on its penetration in the grid's supply mix). Accordingly, the manuscript, including part of the Introduction section, has been thoroughly re-written. We followed the reviewer's suggestion and presented the results and discussions in the following order:

- (i) Results (Displacement efficiency of alternative energy and trans-provincial imported electricity; Displacement efficiency of hydropower and non-hydro alternative energy; Dependence of displacement efficiency on alternative energy penetration) – this section is devoted to discussion of all the important results.
- (ii) Discussion – this section presents explanation of why these results matter or are significant and how they might be explained.
- (iii) Implications for China's Transition towards a Low-carbon Economy and Policy Recommendations – this section mainly presents the implications of the findings on China's alternative energy development and our policy recommendations

Due to space reason, the revised version of these three sections are not shown here, but can be found on *Pages 6-17, Lines 124-382* of the revised manuscript.

Smaller questions/suggestions:

One question: Is the trans-province transported energy all non-fossil-fuel-generated, or is this just referring to all energy that is moved between provinces? I wasn't 100% clear on that, so either way it would help to make that more explicit.

Sorry about the confusion! The trans-province transported electricity refers to all energy that is moved between provinces. Only data on the total quantity of electricity transported between provinces were available, thus we could not differentiate between coal-electricity and alternative

energy in our analysis. We stated in the original manuscript that:

“These results show that electricity imported from other provinces, regardless of its sources of generation, has relatively low use efficiency.” (This sentence has been deleted in the revised manuscript)

To clarify, we added the note of “regardless of the generation sources” to the related sentences when this topic appears for the first time in the Abstract, Introduction, Discussion, and Method sections of the revised manuscript. These sentences are:

“Nationwide, each unit of alternative energy displaced less than one-quarter of a unit of fossil-fuel-generated electricity, while each unit of imported electricity (regardless of the source of generation) displaced ~0.3 unit of fossil-fuel electricity generated locally.” (*Page 2, Lines 26-29*)

“In this work, we used two-way fixed-effects models to determine the efficiency at which alternative energy and trans-provincial imported electricity (regardless of the generation source) displacing the electricity generated by local fossil-fuel-fired plants (i.e., within the provinces) both nationwide and in the six regional grids over the period of 1995-2014.” (*Page 6, Lines 114-118*)

“Thus each unit of electricity imported from other provinces (regardless of the generation source) substituted ~0.3 unit of fossil-fuel electricity generated locally.” (*Page 7, Lines 144-146*)

“The displacement efficiency of trans-provincial imported electricity (regardless of the source of generation), which was not considered in York’s cross-national longitudinal study, was also analyzed, because of the prevalent practice of cross-province transmission in China^{2,39,40}.” (*Page 18, Lines 392-395*)

This paper could also use a close grammatical edit (though there are only a few grammatical mistakes that I noticed).

We thank the reviewer for pointing this out! We have thoroughly and carefully edited the grammar and semantics of the revised manuscript. We made utmost efforts to avoid any grammar and spelling errors, and to prepare a revised manuscript that is clear and compelling.

Overall, I think the paper is a significant contribution to the very important and timely research being done on energy production/the shift towards non-fossil-fuel-based energy/GHG emissions/climate-related public policy, so I wish the authors the best of luck in preparing a strong revision.

We greatly appreciate the reviewer’s very constructive comments and suggestions! Based on the very helpful comments and suggestions from the reviewer, we have thoroughly revised the manuscript by making the following changes:

- (i) We updated the analysis by extending the times series data to 2014, which just became available very recently;
- (ii) We expanded the data analysis by comparing the results in China with those obtained from analysis of global data over the past two decades (i.e., repeating York’s analysis but for a shorter time period to make the time frame comparable with our analysis of China’s data);
- (iii) We analyzed the displacement efficiency of alternative energy and trans-provincial

imported electricity in all the six inter-provincial regional power grids of China, which reveals significant variation depending on local electricity supply/demand and economic conditions;

- (iv) We significantly improved our data interpretation by incorporating classical energy economics literatures;
- (v) We discussed the implications on China's transition towards a low-carbon economy based on the findings of this study.

The very constructive and insightful comments from the reviewers helped us to significantly improve the quality of the revised manuscript. We believe that the changes we have made during the revision help make this study more solid and compelling.

REVIEWERS' COMMENTS:

Reviewer #3 (Remarks to the Author):

The authors put a significant amount of work into this careful and thorough revision and in my opinion successfully addressed all of my comments and suggestions. I think the paper is a strong contribution to the growing literature on the effectiveness of alternative energy production to meet greenhouse gas emissions reductions targets and is suitable for publication at this time.

Reviewer #4 (Remarks to the Author):

With the exception of the second point raised by Reviewer 3 about the repetitiveness of the latter half of the paper, I believe the authors have adequately addressed the concerns raised in the last round of reviews.

Responses to Review Comments

We thank Reviewers #3 and #4 for their careful read and constructive comments on the previous round of revision! Based on the comments from Reviewer #4, we have carefully edited the revised manuscript to eliminate repetitiveness, and to improve its clarity and impact. The following summarizes how we responded to the review comments.

Reviewer #3 (Remarks to the Author):

The authors put a significant amount of work into this careful and thorough revision and in my opinion successfully addressed all of my comments and suggestions. I think the paper is a strong contribution to the growing literature on the effectiveness of alternative energy production to meet greenhouse gas emissions reductions targets and is suitable for publication at this time.

We thank the reviewer for the very positive comment on our work!

Reviewer #4 (Remarks to the Author):

With the exception of the second point raised by Reviewer 3 about the repetitiveness of the latter half of the paper, I believe the authors have adequately addressed the concerns raised in the last round of reviews.

We greatly appreciate the reviewer's comment! In the previous round of revision, we have incorporated new analysis results and thoroughly revised all sections of the manuscript, paying special attention to avoid repetitiveness in the Discussion section.

Based on the reviewer's comment, we thoroughly checked the entire manuscript to fully eliminate repetitiveness, and deleted some less important sentences/discussions. Specifically, the following sentences have been deleted in the revised manuscript:

Introduction section

- "In general, hydropower is the lowest-cost power generation option, whereas some renewables, mainly wind and solar power, have already become, or would soon become cost-competitive with coal-fired generation."

- “Effective and efficient displacement of fossil fuels with the expansion of alternative energy in China is of particular importance for the global low-carbon transition, which is underway and gaining more momentum.”

Discussion section

- “With a long history of development, the highly dispatchable hydropower is among the most cost-effective energy sources and can readily substitute coal-fired generation at supplying baseload⁴.”
- “While alternative energy exhibited no statistically significant displacement effect on fossil-fuel-generated electricity at shares below ~10% in the grid’s supply mix, 50-90% of displacement occurred once its share exceeded this threshold.”
- “That is, alternative energy had little displacement effect on fossil-fuel-generated electricity during the early stage of development. Once it made up more than ~10% of the electricity supply on the grid, alternative energy could have good displacement effect on fossil-fuel-generated electricity, probably due to the scales of economies and the learning effects^{22,24}. Nonetheless, proportional displacement was compromised by the occurrence of the rebound effect, which cancelled out 10-50% of the technologically achievable reduction in fossil-fuel-fired generation.”
- “Similarly, rebound in the consumption of fossil fuels could also occur along with the increased supply of renewables (and nuclear power) at lower costs, partially offsetting the displacement.”

In addition, we also condensed the discussion in some sentences of the revised manuscript. Below are the sentences with significant changes made:

Introduction section

- “As an important growth engine of the global economy, China now accounts for over 16% of the gross domestic product (GDP), 23% of the energy consumption, and 17% of the renewable energy production of the world¹.”

Changed to:

“China accounts for over 16% of the gross domestic product (GDP), 23% of the energy consumption, and 17% of the renewable energy production of the world¹.”

(Page 2, lines 38-39)

- “Driven by rapid industrialization and urbanization over the past three decades, China has become the world’s largest consumer of electricity (5,638 TWh in 2014), with thermal generation accounting for over three-quarters of the total supply².”

Changed to:

“It is also the world’s largest consumer of electricity (5,638 TWh in 2014), with thermal generation accounting for over three-quarters of the total supply².” (*Page 2, lines 39-41*)

- “As a result of the dominance of coal in the primary energy mix, China is the world’s biggest emitter of anthropogenic greenhouse gases (GHGs), and coal-fired power generation contributes largely to the widespread air pollution^{3,5}.”

Changed to:

“As a result, China is the world’s biggest emitter of anthropogenic greenhouse gases (GHGs), and coal-fired power generation contributes largely to the widespread air pollution^{3,5}.” (*Pages 2-3, lines 42-44*)

- “Increasing the production of alternative energy to substitute fossil fuels has been pursued worldwide to improve the sustainability and mitigate the environmental impact of energy use, particularly GHG emissions, over the past decades⁶⁻⁸.”

Changed to:

“Increasing the production of alternative energy to substitute fossil fuels has been pursued worldwide to improve the sustainability and mitigate the environmental impact of energy use⁶⁻⁸.” (*Page 3, lines 46-47*)

- “Meanwhile, although falling behind the United States, France, and Japan in generation capacity, nuclear power in China is expanding at the fastest rate, despite of the nuclear power phase-out in some European countries after the Fukushima nuclear disaster in 2011¹⁰.”

Changed to:

“Meanwhile, nuclear power in China is growing quickly, despite of the phase-out in some European countries after the recent Fukushima nuclear disaster¹⁰.” (*Page 3, lines 54-56*)

Discussion section

- “The results presented above indicate that although far from the proportional displacement desired, alternative energy was utilized two times more efficiently at substituting fossil-fuel-fired electricity generation in China than the global average over the past two decades.”

Changed to:

“The results presented above indicate that alternative energy in China displaced fossil-fuel-fired electricity generation at twice the efficiency of the global average.” (*Page 10, lines 221-222*)

- “This is mainly attributed to the unquenchable thirst for electric power during the

course of fast industrialization and urbanization in China, and the trend is expected to continue in the near future.”

Changed to:

“This is mainly attributed to the unquenchable thirst for electric power during the course of fast industrialization and urbanization in China.” (*Page 10, lines 222-224*)

- “The rapidly rising power demand brought by the strong economic growth and the improving standards of living in China had led to regional and seasonal power shortages in the coastal provinces since the early 2000s⁴. Due to the existence of significant unmet demand, the increased supply of alternative energy would be readily consumed in growing the economy, resulting in relatively high overall efficiency of alternative energy at replacing fossil-fuel-fired generation.”

Changed to:

“Regional and seasonal power shortages had emerged in the coastal provinces of China since the early 2000s⁴, and with the existence of such significant unmet demand, the increased supply of alternative energy would be readily consumed in growing the economy, resulting in relatively high overall efficiency for displacing fossil-fuel-fired generation.” (*Page 11, lines 229-232*)

- “Environmental economists have long contended that improved efficiency, instead of lowering resource consumption, may lead to higher production and consumption (i.e., the Jevons paradox), and proposed that efficiency gains must be paired with conservation policies (e.g., taxation and cap-and-trade), which can reduce consumption by keeping the cost of resource use the same or even higher³⁴⁻³⁸.”

Changed to:

“Environmental economists have long argued that efficiency gains must be paired with conservation policies (e.g., taxation and cap-and-trade) to reduce consumption by keeping the cost of resource use the same or even higher³⁴⁻³⁸.” (*Pages 15-16, lines 339-342*)

There are also some minor word edits, which are not shown here, but can be found on the copy with all changes made to the manuscript tracked.